# Seasonal variation of CaCO₃ saturation state in bottom water of a biological hotspot in the Chukchi Sea, Arctic Ocean

Michiyo Yamamoto-Kawai [1], Takahisa Mifune [1], Takashi Kikuchi [2], Shigeto Nishino[2]

[1]Tokyo University of Marine Science and Technology, Tokyo, 108-8477, Japan

[2] Japan Agency for Marine-Earth Science and Technology, Yokosuka, 237-0061, Japan

*Correspondence to*: Michiyo Yamamoto-Kawai (michiyo@kaiyodai.ac.jp)

**Abstract.** Distribution of calcium carbonate saturation state ($\Omega$) was observed in the Chukchi Sea in autumn 2012 and early summer 2013. $\Omega$ in bottom water ranged from 0.3 to 2.0 for aragonite and from 0.5 to 3.2 for calcite in 2012. In 2013, $\Omega$ in bottom water was 1.1~2.8 for aragonite and 1.7~4.4 for calcite. Aragonite and calcite undersaturation were found in high productivity regions in autumn 2012 but not in early summer 2013. Comparison with other parameters has indicated that biological processes -respiration and photosynthesis- are major factor controlling regional and temporal variability of $\Omega$. From these ship-based observations, we have obtained empirical equations to reconstruct $\Omega$ from temperature, salinity and apparent oxygen utilization. Using two-year-round mooring data and these equations, we have reconstructed seasonal variation of $\Omega$ in bottom water in Hope Valley, a biological hotspot in the southern Chukchi Sea. Estimated $\Omega$ was high in spring and early summer, decreased in later summer, and remained relatively low in winter. Calculations indicated a possibility that bottom water could have been undersaturated for aragonite on an intermittent basis even in the pre-industrial period, and that anthropogenic $CO_2$ has extended the period of aragonite undersaturation to more than two-fold longer by now.

## 1 Introduction

During the last decade, ocean acidification due to uptake of anthropogenic carbon dioxide ($CO_2$) has emerged as an urgent issue in ocean research (e.g., Raven et al., 2005; Orr et al., 2005). Increasing acidity and consequent changes in seawater chemistry are expected to impact marine ecosystem and may threaten some organisms (e.g., Gattuso and Hansson 2011; Branch et al. 2013). Of particular concern is the impact on calcifying organisms, such as coralline algae, pteropods, bivalves, corals etc., because acidification lowers the saturation state ($\Omega$) of calcium carbonate ($CaCO_3$) in seawater which affects the ability of these organisms to produce and maintain their shells or skeletons. In fact, a decrease in $\Omega$ of water can cause enhanced mortality of juvenile shellfish (Green et al., 2009; Talmage and Gobler, 2009) and decreased calcification, growth, development and abundance of calcifiers (Kroeker et al., 2013 and references therein).

The shallow shelf seas of the Arctic Ocean are known to be especially vulnerable to ocean acidification. Cold water dissolves more $CO_2$, large freshwater inputs from rivers and sea ice melt reduce calcium ion concentrations and alkalinity, the buffering capacity of seawater to added $CO_2$ (Salisbury, 2008; Yamamoto-Kawai et al., 2011), and respiration at the bottom of salt

stratified water column accumulates $CO_2$ in bottom water (Bates et al., 2009). Because of these characteristics, both surface and bottom waters of Arctic shelf seas exhibits naturally low $\Omega$ compared to other ocean waters (e.g., Fabry et al., 2009; Mathis et al., 2015; Yamamoto-Kawai et al., 2013). The Chukchi Sea is one of these seas.

Recent studies have found bottom waters already undersaturated with respect to aragonite-type $CaCO_3$ in the Chukchi Sea during summer and autumn (Bates et al., 2009; 2013; Bates, 2015; Mathis et al., 2014). Bates et al. (2013) reported that ~40% of sampled bottom waters during summertime cruises between 2009 and 2011 were aragonite-undersaturated. Some bottom waters were also undersaturated with respect to calcite, a less soluble form of $CaCO_3$ than aragonite. These studies indicate that benthic communities in large areas of this shelf sea has been exposed to bottom waters that are corrosive for their $CaCO_3$ shells and skeletons, at least seasonally. However, full seasonal variation of $\Omega$ is still unrevealed due to the lack of shipboard observations during ice-covered months. Because many benthic organisms have planktonic larval stages, timing and duration of $CaCO_3$ can be critical for their growth and populations.

In the present paper, we show results of ship-based observations in September/October of 2012 and early July of 2013 in the Chukchi Sea. In both years, sea ice concentration decreased to < 80% in the end of May and to <30% in the beginning of June. Therefore, observations were carried out about 3 months and 1 months after the ice retreat in 2012 and 2013, respectively. Because of the difference in season as well as interannual variability in hydrographic conditions, distribution of $\Omega$ was largely different between two observations. By comparing with distributions of physical and biogeochemical parameters, factors controlling $\Omega$ will be discussed. Based on these results, we attempt to reconstruct seasonal variations of $\Omega$ in the bottom water in the southern Chukchi Sea by using data from a two-year-round mooring observation between July 2012 and July 2014.

## 2 Study Area

The Chukchi Sea is a vast and shallow (~50 m) shelf sea north of the Bering Strait (Fig. 1) and is covered by ice for 7 or 8 months of a year. Pacific-origin water enters through the Bering Strait and distributes across the Chukchi Sea. The Pacific water transports heat and freshwater to the Arctic Ocean during summer months. In winter, atmospheric cooling and brine injection from growing sea ice mixes whole water column and modifies the water to be "Pacific winter water" which is characterized by low temperature and relatively high salinity (~33). In coastal polynyas, hypersaline winter water (S> 34) occasionally forms. Even in summertime, the remnant Pacific winter water is often observed at the bottom of the shelf beneath the warm and fresh upper layer.

The Pacific water also transports nutrients to the Chukchi Sea. Nutrient-rich water upwelled onto the Bering shelf are carried northward, promoting very high primary productivity in the sea (e.g., Springer and McRoy, 1993). Using the nutrients, spring phytoplankton bloom occurs immediately after the sea ice retreat or even under sea ice (Fujiwara et al., 2016; Lowry et al., 2014). Relatively high primary production continues to autumn (Wang et al., 2005). Because of the shallow bottom depth and mismatch between seasonal dynamics of phytoplankton and zooplankton, a large proportion of produced organic matter is directly delivered to the seabed (e.g., Grebmeier et al., 2006). As the result, high export rate of organic matter supports very

high benthic biomass that are prey for higher trophic levels such as diving ducks, seals, whales and walruses (Grebmeier et al., 2006; Fabry et al., 2009)..

Exported organic matter is remineralized back to nutrients and $CO_2$ in shelf bottom water and sediments. This accumulates $CO_2$ in bottom water and lowers $\Omega$, to the level of $CaCO_3$ undersaturation in summer/autumn (e.g., 2009). Calcifying bivalves (Fig. 1), amphipods, brittle stars and crabs are dominant spices in benthic community of the Chukchi Sea (Fabry et al.,2009; Blanchard et al., 2013) and are key component of the ecosystem. Ocean acidification, therefore, could have considerable impacts on the ecosystem and biogeochemical cycles of the Chukchi Sea. In addition, because of shallow bottom depth of ~50 m, vertical mixing induced by wind, tide or atmospheric cooling can bring anthropogenic $CO_2$ into bottom water to which benthos are exposed.

## 3 Observation and analysis

Hydrographic data were collected in the Chukchi and Bering Seas during the cruises of R/V Mirai of the Japan Agency for Marin-Earth Science and Technology (JAMSTEC) from 13 September to 4 October in 2012 (Kikuchi et al., 2012) and T/S Oshoro-Maru of Hokkaido University from 3 July to 18 July in 2013 (Hirawake et al., 2013). In the present paper, we use data from stations north of 66°N with bottom depth shallower than 70 m. During both cruises, hydrographic casts were performed using a Sea Bird 9plus CTD to which a carousel water sampler with Niskin bottles was mounted. Seawater samples were collected for total alkalinity (TA) and dissolved inorganic carbon (DIC) as well as complementary data (S: salinity, T: temperature, DO: dissolved oxygen, and nutrients). Samples for TA and DIC were collected according to Dickson et al. (2007). TA was measured using a spectrophotometric system (Yao and Byrne, 1998) for samples from Mirai and an open cell titration system (Dickson et al., 2007) for samples from Oshoro-Maru. Intercomparison of two method in a previous study (Li et al., 2013) showed good agreement of $0.88\pm2.03$ µmol kg$^{-1}$. Also, S-TA relationship between two cruises did not show any obvious offset (Fig. 2). DIC from both cruises were analysed using a coulometer system. Measurements of TA and DIC were calibrated against a certified reference material distributed by A. G. Dickson at Scripps Institute of Oceanography, USA, or General Environmental Technos Co., Japan. The pooled standard deviation (Sp) for duplicate samples was < 2.2 µmol kg$^{-1}$ and < 5.5 µmol kg$^{-1}$ for TA and DIC, respectively. Observations of TA, DIC, nutrients, T, S and pressure were used to calculate $\Omega$ for aragonite ($\Omega$ar) or calcite ($\Omega$ca) and fCO$_2$ (fugacity of $CO_2$) by using the CO2sys program (Lewis and Wallace, 1998) with constants of Mehrbach et al. (1973) refit by Dickson and Millero (1987) for K1 and K2 and Dickson (1990) for KSO$_4$. We have chosen these constants following previous studies in the Arctic Ocean (e.g., Azetsu-Scott et al., 2010; Bates et al., 2009; 2015). However, as evaluated by Azetsu-Scott et al. (2010), the use of these K1 and K2 constants will give lower $\Omega$ than using other constants. The maximum difference of 0.07 in $\Omega$ar was estimated in their study in the Canadian Arctic Archipelago and in Labrador Sea. In the case of our dataset, $\Omega$ will be higher by up to 0.09 for aragonite and 0.14 for calcite when other set of

K1 and K2 listed in Azetsu-Scott et al. (2010) are used. This will be included in error estimates for reconstruction of seasonal variation in $\Omega$.

Bottle DO was determined by Winkler titration following World Ocean Circulation Experiment Hydrographic Program operations and methods (Dickson, 1996) with precision of 0.12 µmol kg$^{-1}$. Nutrient samples were analysed according to the

GO-SHIP Repeat Hydrography Manual (Hydes et al., 2010) with precision of <0.1 µmol kg$^{-1}$ for nitrate, nitrite and phosphate, <0.6 µmol kg$^{-1}$ for silicate, and <0.4 µmol kg$^{-1}$ for ammonium. Cumulative analytical errors in TA, DIC, phosphate and silicate can cause a maximum error of 0.08 in $\Omega$ar and 0.13 in $\Omega$ca.

A mooring system was deployed in Hope Valley from 16 July 2012 to 19 July 2014. The system was first deployed at 67° 42'N, 168° 50'W from 16 July 2012 to 2 October 2012, and then moved slightly to the north at 68° 02'N, 168° 50'W on 3

October 2012. On 20 July 2013, the system was recovered for maintenance and redeployed at the latter location until 19 July 2014. Bottom depth was 52, 59 and 60 m, respectively for each deployment. Sensors for T, S, DO, chlorophyll a and turbidity were equipped on the mooring at 7 m above from the bottom. Data were recorded every hour. T and S data were acquired using a MicroCAT C-T recorder, SBE 37-SM (Sea-Bird Electronics, USA). Maximum drift in sensors over 1 year were 0.002 °C for T and 0.01 for S in pre- and post-calibration comparisons (Nishino et al., 2016). DO sensor used for mooring observation

was an AROW-USB phosphorescent DO sensor (JFE Advantech Co., Ltd., Japan). The sensor was calibrated using oxygen-saturated and anoxic water to determine the linear relationship between them with 2 % accuracy (Nishino et al., 2016). Note that a correction of -69 µmol kg$^{-1}$ applied for the third mooring data in Nishino et al. (2016) was found to be due to an artificial error in conversion of original sensor output to concentration in µmol kg$^{-1}$. With correct conversion, the difference between sensor data and bottle data obtained on 1 September 2013 was reduced from 69 µmol kg$^{-1}$ to 4 µmol kg$^{-1}$. Accordingly, we did

not apply any correction to DO sensor data in the present paper. Details of other sensor and mooring observations are described in Nishino et al. (2016).

## 4 Results and discussions

### 4.1 Ship-based observations

Distributions of $\Omega$ in bottom waters of the Chukchi Sea (Fig. 1) were significantly different between September/October 2012

(hereafter autumn 2012) and July 2013 (hereafter summer 2013). In autumn 2012, $\Omega$ in bottom water ranged from 0.3 to 2.0 for aragonite and from 0.5 to 3.2 for calcite. Aragonite undersaturation ($\Omega$ar < 1; black circles in Fig. 1) was observed at stations off Pt. Barrow, in Hope Valley, in Bering Strait, and near the northern continental slope. The lowest $\Omega$ar was observed on 3 October 2012 in Hope Valley at 68ºN, in the dome-like structure of bottom water with low T, high S and low DO (Nishino et al., 2016) with high TA, high DIC and high fCO$_2$ (Fig. 3). Calcite undersaturation was also found in this dome (Figs. 1 and

3). Nishino et al. (2016) describe the dome-like structure as a common feature found in this region associated with Hope Valley topographic depression, although water properties can differ between years and seasons. Note that stations along 168 °W in Hope Valley were visited twice on 13-17 September and 3-4 October in 2012 and difference in $\Omega$ values of bottom water

between the two visits (< 0.2) were smaller than differences between stations inside and outside of the dome-like structure (~0.7).

In 2013, $\Omega$ in bottom water was 1.1~2.8 for aragonite and 1.7~4.4 for calcite: all of observed waters were oversaturated with respect to aragonite and calcite (Figs. 1 and 4). Although stations in Hope Valley again showed a dome-like feature in T, DO, and $fCO_2$, they were not as prominent as in 2012 (Fig. 4). $\Omega$ in bottom waters at these stations were slightly lower than stations north or south but still well oversaturated with respect to $CaCO_3$ ($\Omega$ar =1.7 ~1.9; $\Omega$ca =2.8 ~3.0). Waters with lower $\Omega$ar of 1.1~1.2 and $\Omega$ca of 1.8 ~1.9 were found in northern stations at around 71ºN where T and DO were lower and $fCO_2$ was higher than in the south.

In both years, low $\Omega$ was observed in bottom waters with low T, low DO and high S. Relationships between $\Omega$ar and T, S or apparent oxygen utilization (AOU; difference between saturation and observed concentrations of DO) are shown in Fig. 5. $\Omega$ar showed the highest correlation with AOU and data from two cruises were distributed on a line in the $\Omega$ar-AOU diagram (Fig. 5). AOU is a measure of how much oxygen has been consumed by respiration and decomposition of organic matter. Accordingly, high AOU corresponds to high $fCO_2$ (Fig. 6) and therefore low $\Omega$ (Fig. 5). Negative AOU value is a sign of photosynthesis which produces DO, consumes $CO_2$ and increases $\Omega$. As shown in Fig. 6, biogeochemical conditions between two ship-based observations were different: autumn 2012 was under very strong influence of remineralization whereas summer 2013 was influenced of photosynthesis in early summer 2013.

$\Omega$ar increased with increasing T and their relationships for two cruises were similar in slope but different in intercept (Fig. 5). $\Omega$ar also increased with decreasing S in bottom waters, whereas $\Omega$ar decreased with decreasing S in upper waters (Fig. 5). The latter is explained by mixing with freshwater, which lowers calcium ion concentration and alkalinity to decrease $\Omega$ar as indicated by previous studies (Salisbury et al., 2008; Yamamoto-Kawai et al., 2009). In fact, aragonite undersaturation were observed in very low S surface waters in 2012 (Fig. 5). The opposite relationship in bottom waters is probably because higher bottom S creates stronger stratification of water column that prevent release of $CO_2$ produced by respiration at depth. These results show that variations in $\Omega$ in the Chukchi Sea bottom water is controlled largely by organic matter remineralization with minor contributions of T and S. In fact, low $\Omega$ waters were observed in regions off Pt. Barrow and Hope Valley, known as biological "hotspots" in the Chukchi Sea, characterized by high primary productivity, high export flux of organic matter to depth, high respiration rate in sediment community, and high benthic biomass (Grebmeier et al., 2006; Nishino et al., 2016).

In autumn 2012, $\Omega$ was much lower than in summer 2013, not only in bottom water but for the whole water column (Fig. 3). Nishino et al. (2016) compared hydrographic conditions in this area in late summer of 2004, 2008, 2010, 2012 and 2013, and found that 2012 was unusual year with strong stratification, due to an input of sea ice meltwater, and with remarkably low DO concentration at depth. Strong stratification prevents ventilation and accumulates more $CO_2$ in the bottom water. This explains high AOU, high $fCO_2$ and low $\Omega$ in bottom water in autumn 2012 (Fig. 3). Seasonal variation in $\Omega$ should also be a cause of the difference between two observations as described in the following sections. In upper layers of the water column, lower $\Omega$ in 2012 relative to S and T than in 2013 (Fig. 5) might be due to low photosynthetic activity associated with stronger stratification and to mixing with sea ice meltwater (Nishino et al., 2016). In addition, input of sea ice meltwater itself lowered

Ω in surface waters in 2012 as evident in Fig. 5. The fact that ranges of T and S for bottom waters were not significantly different between two cruises (Fig. 5) indicates that the accumulation of more $CO_2$ produced by respiration was the major cause of the much lower $\Omega_{ar}$ in autumn 2012 than summer 2013.

## 4.2 Mooring observations

Mooring data revealed seasonal variation in T, S, DO and AOU in bottom water in Hope Valley where aragonite and calcite undersaturations were observed in autumn 2012 (Fig. 7). Ship-based observations were made near the mooring site (~68°N) on 14 September 2012, 3 October 2012 and 16 July 2013 and a comparison with sensor data is summarized in Table 1, together with observations on 1 September 2013 by Nishino et al. (2016). 24-hr mean data are shown in Table 1 for sensor data with standard deviations. Differences in Table 1 reflect not only instrumental errors but also inhomogeneity of water properties at the observation sites. This is likely a reason of relatively large differences found in all T, S and DO data on 15 July 2013 when ship-based observation was made 31 km away from the mooring site. Other than this, two observations agreed within 0.2 °C in T, 0.15 in S and 11 µmol kg$^{-1}$ in DO.

T, S and DO (AOU) ranged from -1.92 to 2.73 °C, 30.41 to 35.49, and 62.7 to 416.4 (-32.7 to 294.9) µmol kg$^{-1}$, respectively. Note that ship-based observations were made when DO was at higher and lower parts of seasonal variation. During summer/autumn months, T, S and DO showed large fluctuations. Because all three parameters changed simultaneously, this is likely due to changes in water current which carries water masses with different characteristics from surrounding areas. This also indicates horizontal inhomogeneity of bottom water properties during summer/autumn.

T was high in summer/autumn months, decreased from October to December due to atmospheric cooling, and remained at a near freezing temperature (~ -1.8 °C) from January to May. Summer/autumn T was lower in 2012 than in 2013. Stronger stratification of water column due to a meltwater input in 2012 (Nishino et al., 2016) may be the cause of this difference. This is also consistent with higher S in bottom water in 2012 than in 2013. Mean S for the whole observation period was 32.4. In both years, freshening was observed at the beginning of the cooling period, because of mixing of low-S upper waters into the bottom layer by vertical convection (Woodgate et al., 2005). Another freshening was also found at the beginning of warming periods due to melting of sea ice (Woodgate et al., 2005). In winter of 2013, S increased rapidly in February to as high as 35.49 and then decreased sharply in March. This suggests an advection of water mass from active ice formation area, such as coastal polynya region where brine rejection from freezing seawater increases S of bottom water to form "hypersaline water" (Weingartner et al., 1998). Such event was not observed in the following winter.

DO increased and reached supersaturation (negative AOU) in May and June (Fig. 7), accompanied by a sharp increase in chlorophyll-a concentration (Nishino et al., 2016), indicating effects of oxygen production by photosynthesis even in the bottom water. DO then decreased from ~300 µmol kg$^{-1}$ in July to ~125 µmol kg$^{-1}$ in September 2012 and to ~225 µmol kg$^{-1}$ in September 2013 (Fig. 7). DO was remained low until the onset of winter convection in October/November. As mentioned in the previous section, DO in bottom water was unusually low in autumn 2012, due to strong stratification that prevented ventilation of bottom water. During winter, DO was relatively stable at ~325 µmol kg$^{-1}$. This concentration corresponds to ~60

μmol kg$^{-1}$ in AOU and ~85 % in saturation level even in the hypersaline water that is formed in contact with atmosphere. This suggests that positive AOU (DO undersaturation) in winter is not due to insufficient gas exchange. Continued sediment oxygen uptake is a possible reason for the undersaturation. Previous studies in shallow Arctic seas have found that sediment oxygen uptake rate is regulated by the availability of organic matter and macrofaunal biomass (Grebmeier and McRoy, 1989; Rysgaard et a l., 1998; Grant et al., 2002; Clough et al., 2005). Accordingly, oxygen uptake rate has a seasonal variation and is low in winter prior to initiation of biological production in spring (Cooper et al., 2002; Grant et al., 2002). Nevertheless, winter sediment oxygen uptake rate is not zero but is about half of that in summer in coastal area north of Pt. Barrow, Alaska (Devol et al., 1997), in Young Sound in Northeast Greenland (Rysgaard et al., 1998) and in Resolute Bay in Canadian high Arctic (Welch et al., 1997). In our observations, mean AOU in mid-winter (February to April) was 1/3 ~ 1/2 of that in October. Therefore we consider that positive AOU was maintained by benthic respiration during winter.

The analysis of mooring data indicates that our ship-based observation in summer 2013 sampled bottom water that was under an influence of photosynthesis. In autumn 2012, on the other hand, we have observed bottom water that was largely affected by organic matter decomposition. This explains the differences in AOU and Ω in bottom water between two ship-based observations in September/October 2012 and July 2013.

## 4.3 Regression analysis

Based on ship-based and mooring observations, we try to reconstruct seasonal evolution of Ω in the bottom water of Hope Valley. Previous studies have used multiple linear regression models to robustly determine carbonate parameters such as DIC, TA, Ω, and pH from observations of T, S, DO or nutrients (Juranek et al., 2009; Alin et al., 2012). These empirical equations were successfully used to reconstruct seasonal and interannual cycles, as well as high-frequency variability in short-time scales (Juranek et al., 2009; Alin et al., 2012; Leinweber and Gruber, 2013; Evans et al., 2013). In the present study, we employ a similar approach to estimate Ω. We use observations from two cruses to determine regression equations for DIC and TA using T, S and AOU as input parameters. The goodness of the fit was assessed by correlation coefficients ($r^2$) and root mean square error (RMSE). AOU, not DO itself, is used as a measure of biological process because DO concentration changes with T and S. We then calculate Ω from estimated TA and DIC with observed T, S and pressure, rather than directly estimate Ω from input parameters. In this way, we can take into account the effects of T, S and pressure on solubility of $CaCO_3$ (Mucci, 1983; Millero 1995).

DIC is controlled by physical (solubility and gas exchange) and biological processes (photosynthesis and respiration) and therefore is can be expressed as a function of T, S and AOU. TA in the study area is a function of S, as it is determined by mixing of Pacific-origin seawater with freshwater and formation of sea ice (brine rejection increase both S and TA of the underlying water). A bloom of calcifying primary producer can cause a drawdown of TA (Murata and Takizawa, 2002). However, neither bloom of calcifies nor TA drawdown in S-TA diagram was observed during our observations (Fig. 2). Mixing between sea water and freshwater sources with different TA (high in river and low in sea ice meltwater and precipitation;

Yamamoto-Kawai et al., 2005) is evident in the S-TA diagram but only in surface waters with S < 31 (Fig. 2). Accordingly, we used T, S and AOU to estimate DIC (DICest) and S to estimate TA (TAest) only for waters with S>31.

Regression equations obtained for DIC and TA are:

$$\text{DICest} = 1.06 \times \text{AOU} - 17.03 \times T + 41.54 \times S + 743.94 \quad (r^2=0.96, \text{RMSE}=24.46, n=184) \qquad (1)$$

and

$$\text{TAest} = 59.23 \times S + 370.34 \quad (r^2=0.83, \text{RMSE}=14.03, n=184) \qquad (2).$$

Although biological activities can change TA to a small extent by adding or removing nitrate or ammonium, the inclusion of AOU did not improve the regression model ($r^2=0.83$, RMSE=14.06). Inclusion or replacement of proxies of other biological processes, such as nutrients or chlorophyll a concentration did not significantly improve the estimate of DIC and TA.

Fig.8a shows a linear correlation between observations ($\Omega$obs) and estimations ($\Omega$est) with $r^2 =0.94$ for aragonite ($r^2 =0.94$ calcite, not shown). Larger differences between two values were found in surface waters with $\Omega$est values higher than 2.5 for aragonite and 4.0 for calcite (Fig. 8a) and with high temperature (>6°C). This might be due to rapid warming at the surface that could cause temporal decoupling of oxygen and carbon because of differences in temperature dependence of solubility and in gas exchange rate. Including these samples, RMSE was 0.17 for $\Omega$ar and 0.27 for $\Omega$ca (n = 184)(Figure 8a).

In order to evaluate the regression equations (1) and (2), we have applied these to independent data from R/V Mirai cruises in the Chukchi Sea in 2000, 2002, 2006, 2009 and 2010, downloaded from the website of Japan Agency for Marine-Earth science and Technology (JAMSTEC). These cruise observations were carried out in August, September or October. In our study area (latitude> 66°N, bottom depth < 70 m), total 127 set of discrete bottle sample data of DIC, TA and DO, together with sensor data of T and S (>31)were found. For each dataset, we have calculated DICest and TAest from T, S and AOU using the equations (1) and (2). AOU was calculated from discrete bottle DO data. DICest and TAest agreed with observed bottle data of DIC and TA with $r^2=0.96$ and 0.83, respectively. $\Omega$ calculated from DICest and TAest ($\Omega$est) was correlated with observed $\Omega$ ($\Omega$obs) with a regression equation of $\Omega$arest = 1.15 × $\Omega$arobs - 0.06 ($r^2 = 0.78$, RMSE=0.36, Fig. 8b) and $\Omega$caest = 1.14 × $\Omega$caobs - 0.06 ($r^2 = 0.78$, RMSE=0.57). We regard RMSEs estimated in this evaluation represent cumulative errors in sampling, sample analysis, regression analysis, and application of equations to other years with different hydrographic and biogeochemical conditions. In addition to this, a possible systematic error caused by the choice of K1and K2 constants as mentioned in section 3 should be considered. Therefore, we present $\Omega$est with a range of $\pm$RMSE+0.09 for aragonite and $\pm$ RMSE+0.14 for calcite in the following section (-0.36 to +0.45 for $\Omega$arest and -0.57 to +0.71 for $\Omega$caest).

Cross et al. (2013) pointed out the possibility of shallow-water $CaCO_3$ mineral dissolution, which could cause an increase in TA by 36 umol kg$^{-1}$ in the northern Bering Sea where aragonite undersaturation lasts for 5 months from spring to autumn. In our study area, an increase in TA was also found in some bottom waters. However, this was likely due to brine injection, rather than mineral dissolution. This was suggested by a comparison of S, TA and oxygen isotope ratio of water ($\delta^{18}O$) observed in cruises of R/V Mirai in 2000, 2002, 2009, 2010 and 2012 in the Chukchi Sea. The increase in TA was correlated with brine content, estimated from $\delta^{18}O$ as shown in Yamamoto-Kawai et al. (2005), rather than with $\Omega$ of the water. Therefore, we consider that effect of mineral dissolution is insignificant in waters discussed in our analysis.

### 4.4 Seasonal variation of $\Omega$ in Hope Valley bottom water

Based on the analysis in the section 4.3, the regression equations were applied to the mooring data in Hope Valley (excluding data with S < 31 observed intermittently between 8 and 21 November 2013).

The reconstructed $\Omega$ is shown in Fig. 9 with range of -0.36 and +0.45 for aragonite, and -0.57 and +0.71 for calcite. For the whole period, $\Omega$est ranged from 0.2 to 2.1 for aragonite and from 0.3 to 3.4 for calcite (Fig. 9). It was shown that our ship-based observations in autumn 2012 and summer 2013 have captured low and high $\Omega$ periods, respectively. Seasonal variation of $\Omega$ mirrors that of DO, low in autumn due to stratification and respiration and high in spring and early summer due to photosynthesis. In the unusually stratified autumn 2012, bottom water $\Omega$ was ~0.3 for aragonite and ~0.5 for calcite. In 2013,

$\Omega$ decreased after our ship-based observation in July, and intermittent aragonite undersaturation was predicted in August, September and October, although $\Omega$ was higher than in same months in 2012. At the beginning of cooling and convection period in November/December 2012 and October/November 2013, ventilation of bottom water increased DO and $\Omega$. Then, $\Omega$ in bottom water was remained low during winter until the initiation of photosynthesis in May. Low $\Omega$ in winter is likely due to continued respiration by benthic organisms as suggested by positive AOU. Although equations obtained from

summer/autumn data were used to estimate winter $\Omega$, we presume this is acceptable because summer/autumn bottom water is a remnant of winter water that was modified by remineralization of organic matter after spring. If remineralization quotient of DO and DIC is held relatively constant over the course of the year as observed in Young Sound ($\Delta$DIC/$\Delta$DO = 0.8~1.1, Rysgaard et al., 1998), the summer/autumn relationship between DIC (and TA) and T, S, AOU could be applicable to winter data. This assumption should be verified by direct winter observation by ship-based sampling, chemical sensors, or automatic

water samplers in the future.

From spring to autumn, predicted large temporal variation in $\Omega$ suggests inhomogeneous distribution of undersaturated waters during this period. In winter, in contrast, variability in $\Omega$ is relatively small. This suggests that low $\Omega$ is a widespread feature during winter. An exception is the hypersaline water that could have high calcium ion concentration and alkalinity concentration to result in high $\Omega$. However, S of this water is out of the range of ship-based observations used for multiple

linear regression analysis, and thus $\Omega$ of this water is not very reliable.

 Reconstructed $\Omega$ suggests frequent occurrence of aragonite undersaturation in the bottom water of the Chukchi Sea, not only in summer/autumn but also in winter months. In previous studies, continuous aragonite undersaturation has been observed in bottom waters in Bering and Chukchi Sea but in limited seasons. Cross et al. (2013) reported persistent aragonite undersaturation of bottom water of the northern Bering Sea shelf for at least 5 months from mid-April to early October in 2009.

Mathis et al. (2014) indicated sustained bottom aragonite undersaturation on the southern Bering Sea shelf for at least of 4 months from mid-June to early October in 2011. In the Northern Chukchi Sea around 71.5°N and 165°W, Mathis and Questel (2013) observed seasonal changes in $\Omega$ar and reported that bottom water became partially undersaturated in September and broadly undersaturated in October in 2010, with a lowest $\Omega$ar value of ~0.7. The present study is the first to estimate year-

round variability of $\Omega$ in the bottom water of the Chukchi Sea. For the first (from 16 July 2012 to 16 July 2013) and second (from 16 July 2013 to 16 July 2014) full-year mooring observations, integrated period of aragonite undersaturation was counted to be 8.6 months for the first year, and 7.5 months for the second year. For calcite, undersaturation was not as frequent as aragonite (Fig. 9) but suggested not only for autumn 2012 but also in 2013 on an intermittent basis. Integrated period of calcite

undersaturation was 3.1 months and 0.8 months for first and second year, respectively. Considering that the mooring site is located in a biological hotspot where the lowest $\Omega$ was observed in autumn (Figs. 1 and 3), total time of undersaturation estimated here is likely at a maximum within the Chukchi Sea.

## 4.5 Anthropogenic impact on $\Omega$

In order to roughly quantify the effect of anthropogenic $CO_2$ on timing and duration of $CaCO_3$ undersaturation in our 2-year
time series of $\Omega$, we have estimated $\Omega$ in two cases: 1) preindustrial period case with atmospheric partial pressure of $CO_2$ ($pCO_2$) of 280 ppm, and 2) future case with $pCO_2$ of 650 ppm. Following previous studies (Gruber et al., 1996; Sabine et al., 1999; Yamamoto-Kawai et al., 2013; 2015), DIC concentration in a year t is expressed as:

$DICt = DICEQt\text{-}0 + (\Delta diseq + \Delta bio),$

where DICEQt-0 is DIC of seawater in equilibrium with the atmospheric $CO_2$ in the year t-0 when the water parcel was last in
contact with the atmosphere, $\Delta diseq$ and $\Delta bio$ represents air-sea equilibrium at the surface and change in DIC due to biological activity. In case of the shallow Chukchi Sea shelf, t-0 can be assumed to be the observed year (t-obs) and therefore the sum of the last two terms can be calculated by comparing DIC and DICEQ for the year t-obs. In this study, we used DICest calculated using equation (1) with mooring data as DIC in the year t-obs. For calculation of DICEQt-0, we have used $pCO_2$ of 380 ppm, with TAest, T and S from mooring data. Then, assuming that $\Delta diseq$ and $\Delta bio$ do not change with time, DIC in any year t can
be estimated by using atmospheric $pCO_2$ at the year t. We have estimated $\Omega$ with DICt for the pre-industrial period when $pCO_2$ was 280 ppm, and for future when $pCO_2$ reaches 650 ppm (50 years later in the high $CO_2$ emission scenario (RCP8.5), and end of the century in a moderate scenario (RCP6) (IPCC, 2013).

Fig. 10 shows time-series of $\Omega$ for two cases. For the preindustrial case, $\Omega$ ranged from 0.2 to 2.6 for aragonite and from 0.4 to 4.1 for calcite. For the future case, $\Omega$ ranged from 0.2 to 1.5 for aragonite and from 0.2 to 2.4 for calcite. Caveat here is that
our calculation is based on an assumption that terms $\Delta diseq$ and $\Delta bio$ have not changed since pre-industrial period, and therefore provides only very rough estimates. As biological processes are the major factor changing DIC in bottom water, changes in $\Delta bio$ can cause significant error in estimated $\Omega$ for the past and the future. For example, if biological production and subsequent remineralization at depth is lower in the past or in the future, $\Omega$ should be higher than shown in Fig. 10. At the moment, unfortunately, trends in productivity in the southern Chukchi Sea are still under debate. Lee et al. (2007) and Yun et
al. (2014) show that the primary production rate in recent years are lower than previous estimates made in 1990s. From chlorophyll a analysis, Grebmeier (2012) and Arrigo and van Dijken (2011) have suggested an increase in primary productivity in 2000s in the Chukchi Sea. Grebmeier et al. (2015) also showed an increase in benthic biomass from 1970s to 2010 followed by a decline between 2010 and 2012 in our study area. To get a rough idea, we have calculated $\Omega$ for pre-industrial case with

($\Delta$diseq + $\Delta$bio) term half of that at present. With half ($\Delta$diseq + $\Delta$bio), $\Omega$ was estimated to range from 0.6 to 2.2 for aragonite and from 1.0 to 3.5 for calcite in pre-industrial case. This means that $CaCO_3$ undersaturation might have occurred, at least for aragonite, even with the productivity much lower than that occurring today and without perturbation by anthropogenic $CO_2$. This may be the case only in our study site, because previous studies in the Chukchi Sea have suggested that undersaturation in bottom water is a recent phenomenon caused by anthropogenic $CO_2$ (Bates et al., 2009; 2013; Mathis and Questel, 2013).

In terms of duration, period of aragonite (calcite) undersaturation was estimated to be 3.9 (2.6) and 1.7 (0.3) months in the first and second year, respectively, in the pre-industrial period case. By comparing with original estimate of 7.5-8.6 months, it was suggested that the period of aragonite undersaturation has been significantly extended by an input of anthropogenic $CO_2$ by now. In the future case with atmospheric $pCO_2$ of 650 ppm, the period of undersaturation is estimated to increase further to >11 months for aragonite and half a year for calcite. This analysis indicates that anthropogenic $CO_2$ has a significant impact on duration of $CaCO_3$ undersaturation in the bottom water even though seasonal and interannual variations of $\Omega$ is mainly controlled by biological processes.

## 5 Summary and conclusions

Spatial distribution of $\Omega$ in bottom water of the Chukchi Sea was observed in autumn 2012 and early summer 2013. Both aragonite and calcite undersaturation was observed in highly productive regions, including Hope Valley, but only in 2012. Comparison with AOU, T and S showed that organic matter remineralization is the major factor to lower $\Omega$ of bottom water in the Chukchi Sea, with minor but significant control of T and S.

Mooring observations of AOU, T and S for two years in Hope Valley (Nishino et al., 2016) showed that our ship-based observations captured conditions under very strong influence of remineralization in autumn 2012, and under an influence of photosynthesis in early summer 2013. This explains large difference in $\Omega$ between two cruises.

Using cruise observations, we have obtained empirical equations to reconstruct $\Omega$ from data of T, S and AOU and applied them to 2-year round mooring data in Hope Valley. Reconstructed variation of $\Omega$ in bottom water suggested frequent undersaturation for both aragonite and calcite, not only in 2012 but also in 2013. The period of aragonite undersaturation could be more than 7.5 month (60 %) of a year. Such frequent aragonite undersaturation may be harmful for benthic calcifiers who rely on a planktonic early life stages with shells composed of aragonite. Calculations suggest that bottom water in the biological hotspot could have been subject to aragonite undersaturation on an intermittent basis even in the pre-industrial period. It was also suggested that anthropogenic $CO_2$ has significantly extended the period aragonite undersaturation to more than two-fold longer by now. With increased atmospheric $pCO_2$, the period of aragonite undersaturation will extend further. Clearly, anthropogenic $CO_2$ has significant impact on duration of $CaCO_3$ undersaturation in the bottom water even though seasonal and interannual variations in $\Omega$ is controlled by biological processes. We should note that this study is the first attempt to reconstruct seasonal variation of $\Omega$ using a method that has not been confirmed to work in Arctic shelf seas where seasonal changes in

biological activity are extremely large. Direct observation of carbonate parameters in winter by using sensors or water sampler is desired to confirm our results.

It has been revealed that $CaCO_3$ undersaturation has negative impacts on calcifying organisms (e.g., Kroeker et al., 2013). Therefore, continuous occurrence of undersaturation since pre-industrial period in the southern Chukchi Sea is not consistent

with the fact that bivalves are dominant in benthic community in this area (Grebmeier, 2012; Grebmeier et al., 2015). In fact, when we collected benthic organisms by using a dredge during the cruise of T/S Oshoro-maru in 2013, many bivalves were found in Hope Valley hot-spot area, both well-grown adults and small young individuals (Fig. 1). This may suggest tolerance of these bivalves to $CaCO_3$ undersaturation with protection mechanisms such as external organic layer (Ries et al., 2009), companion of energetic cost of calcification by abundant supply of food (Wood et al., 2008), migration, or mismatch in timing

of their planktonic and settling stages and occurrence of $CaCO_3$ undersaturation in surrounding water. With rapidly increasing anthropogenic $CO_2$ in recent and future years, quantification of the responses of local calcifying organisms to low $\Omega$ is an urgent issue for the future study. The Chukchi Sea, already undergoing $CaCO_3$ undersaturation, should provide a research field to assess vulnerability and resilience of organisms to ocean acidification, or to find direct evidence of consequences of ocean acidification in Arctic seas.

**Acknowledgements**

This work was done as a part of the "Arctic Climate Change Research Project" within the framework of the Green Network of Excellence (GRENE) Program and the "Arctic Challenge for Sustainability project (ArCS)" funded by the Ministry of Education, Culture, Sports, Science and Technology-Japan (MEXT). We thank captains, officers and crews of R/V Mirai,

Japan Agency for Marine-Earth Science and Technology (JAMSTEC), T/S Oshoro-Maru, Hokkaido University, and CCGS S. W. Laurier for their help in sampling and mooring operations. Data from these cruises are, or will be, available at website of JAMSTEC and/or Arctic Data archive System (ADS), Japan. Also used in this study are data acquired during the MR00-K06, MR02-K05, MR06-04, MR09-03, MR10-05 and MR12-E03 cruise of R/V Mirai, downloaded from the JAMSTEC website. Some figures in this paper were illustrated using Ocean Data View software (R. Schlitzer, 2008, available at http://odv/awi.de/).

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

**Table 1.** Comparison of of depth (D), temperature (T), salinity (S) and dissolved oxygen (DO) between mooring and ship-based observations collected at the nearest location and time to the mooring data acquisition. Mooring data are 24-hours mean with standard deviation (S.D.). Distances between mooring site and ship-based sampling site are also indicated in km.

| Date & site distance | Parameter | Mooring 24-hr mean | | S.D. | Ship-based | difference |
|---|---|---|---|---|---|---|
| 09/14/2012 | D [m] | 44.56 | ± | 0.08 | 41.80 | −2.76 |
| 23 km | T [°C] | 1.59 | ± | 0.03 | 1.40 | −0.19 |
| | S | 32.22 | ± | 0.01 | 32.26 | 0.05 |
| | DO [μmol kg$^{-1}$] | 284.76 | ± | 2.50 | 279.80 | −4.96 |
| | | | | | | |
| 10/03/2012 | D [m] | 54.03 | ± | 0.04 | 51.50 | −2.53 |
| 5 km | T [°C] | 0.69 | ± | 0.13 | 0.80 | 0.11 |
| | S | 33.01 | ± | 0.02 | 32.96 | −0.05 |
| | DO [μmol kg$^{-1}$] | 102.89 | ± | 4.40 | 113.65 | 10.76 |
| | | | | | | |
| 07/16/2013 | D [m] | 53.99 | ± | 0.03 | 51.53 | −2.46 |
| 31 km | T [°C] | 1.14 | ± | 0.08 | 0.79 | −0.35 |
| | S | 32.08 | ± | 0.03 | 32.28 | 0.20 |
| | DO [μmol kg$^{-1}$] | 316.32 | ± | 7.28 | 350.40 | 34.07 |
| | | | | | | |
| 09/01/2013 | D [m] | 52.78 | ± | 0.05 | 52.50 | −0.28 |
| 4 km | T [°C] | 2.31 | ± | 0.10 | 2.29 | −0.03 |
| | S | 32.26 | ± | 0.05 | 32.41 | 0.15 |
| | DO [μmol kg$^{-1}$] | 267.23 | ± | 8.20 | 270.92 | 3.69 |

**Figure captions**

**Figure 1.** Distribution of $\Omega$ar (a and c) and $\Omega$ca (b and d) in bottom water in September/October 2012 (a and b) and July 2013 (c and d). Circled stations were undersaturated with $CaCO_3$ minerals. Stations in red polygon were used in Fig. 3. White arrows indicate mooring sites. An insert photo shows bivalves collected by a dredge trawl at a station marked with a star in July 2013.

**Figure 2.** Relationship between salinity and total alkalinity (TA, $\mu$mol kg$^{-1}$) observed during cruises in 2012 and 2013.

**Figure 3.** Vertical sections of (a) salinity, (b) temperature, (c) dissolved oxygen (DO), (d) Dissolved Inorganic Carbon (DIC), (e) Total Alkalinity (TA), (f) fCO$_2$, (g) $\Omega$ar and (e) $\Omega$ca in September/October 2012. See Fig. 1 for locations.

**Figure 4.** Same as Figure 4 but in July 2013. See Fig. 1 for locations.

**Figure 5.** Relationships between (a) $\Omega$ar and temperature, (b) $\Omega$ar and salinity, and (c) $\Omega$ar and apparent oxygen utilization (AOU, $\mu$mol kg$^{-1}$).

**Figure 6.** Relationships between AOU ($\mu$mol kg-1) and fCO$_2$ ($\mu$atm).

**Figure 7.** Time series of salinity (top), temperature (middle; °C), and dissolved oxygen and AOU (bottom; $\mu$mol kg-1). Red symbols indicate ship-based observations. In the bottom panel, dots and squares are ship-based data of dissolved oxygen and AOU, respectively.

**Figure 8.** Comparison between $\Omega$ar estimated from T, S, and AOU data using equation (1) and (2) ($\Omega$ar (est)), and $\Omega$ar estimated from bottle DIC and TA ($\Omega$ar (obs)) for ship-based-cruises (a) in 2012 and 2013, and (b) in 2000, 2002, 2006, 2009 and 2010.

**Figure 9.** Time series of $\Omega$ar (top) and $\Omega$ca (bottom) reconstructed from mooring data of T, S and AOU (a black line). Red symbols indicate ship-based observations. Gray linesindicate range of -0.36 to +0.45 for $\Omega$ar and -0.57 to +0.71 for $\Omega$ca (see text).

**Figrue 10.** Time series of $\Omega$ar (top) and $\Omega$ca (bottom) for cases when atmospheric $CO_2$ concentration was 280 ppm (blue; pre-industrial period) or 650 ppm (red). See text for details.

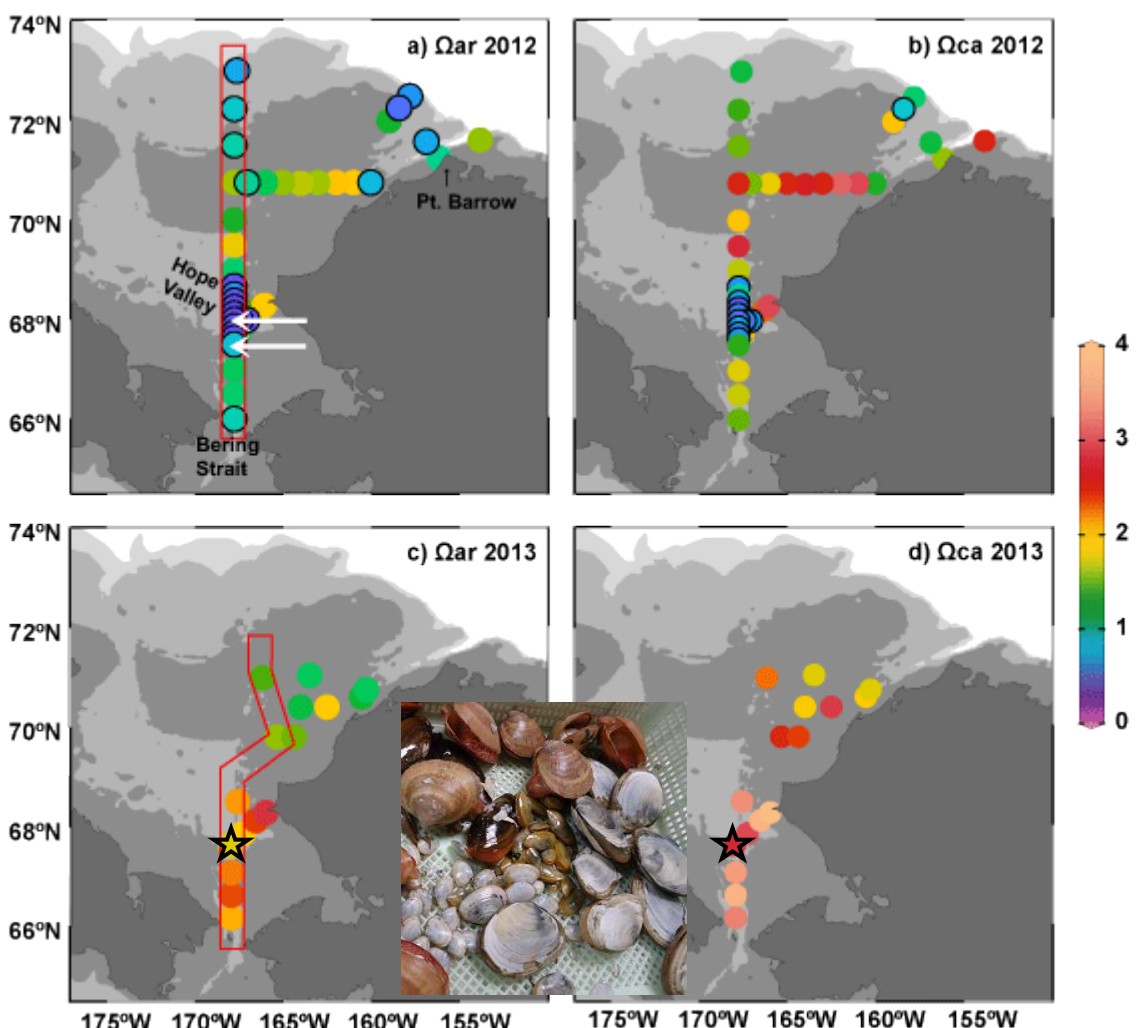

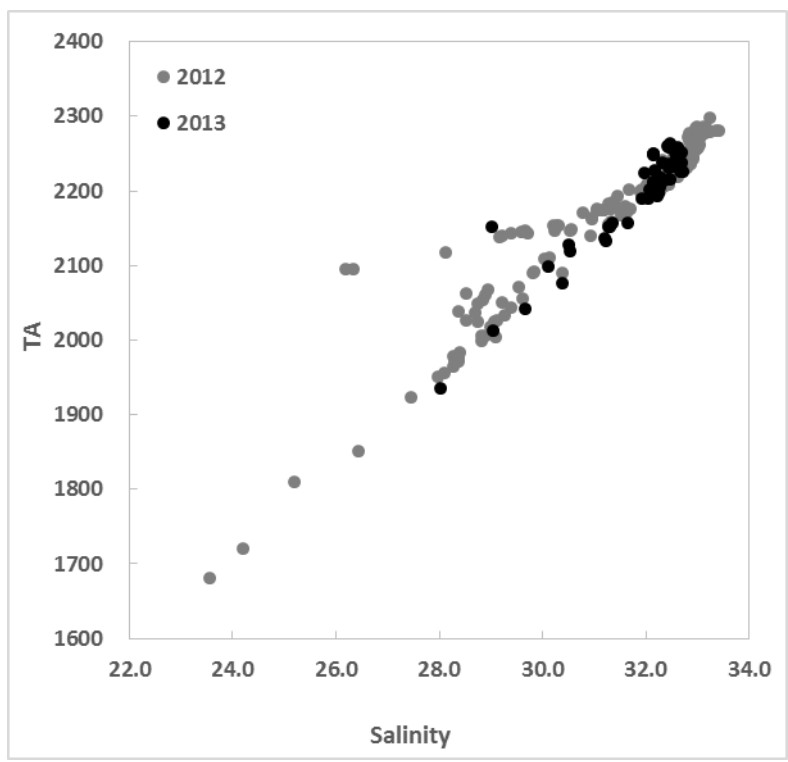

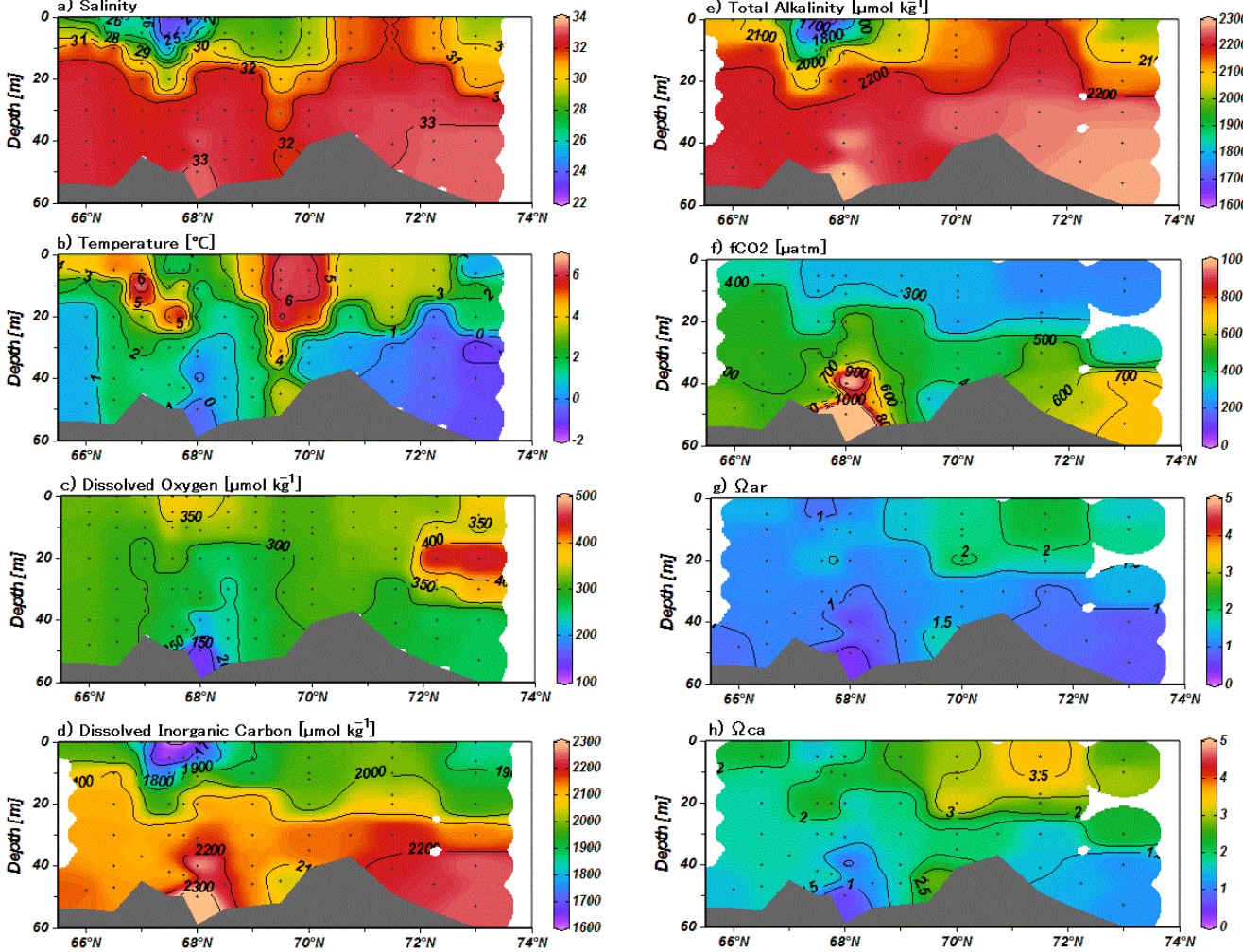

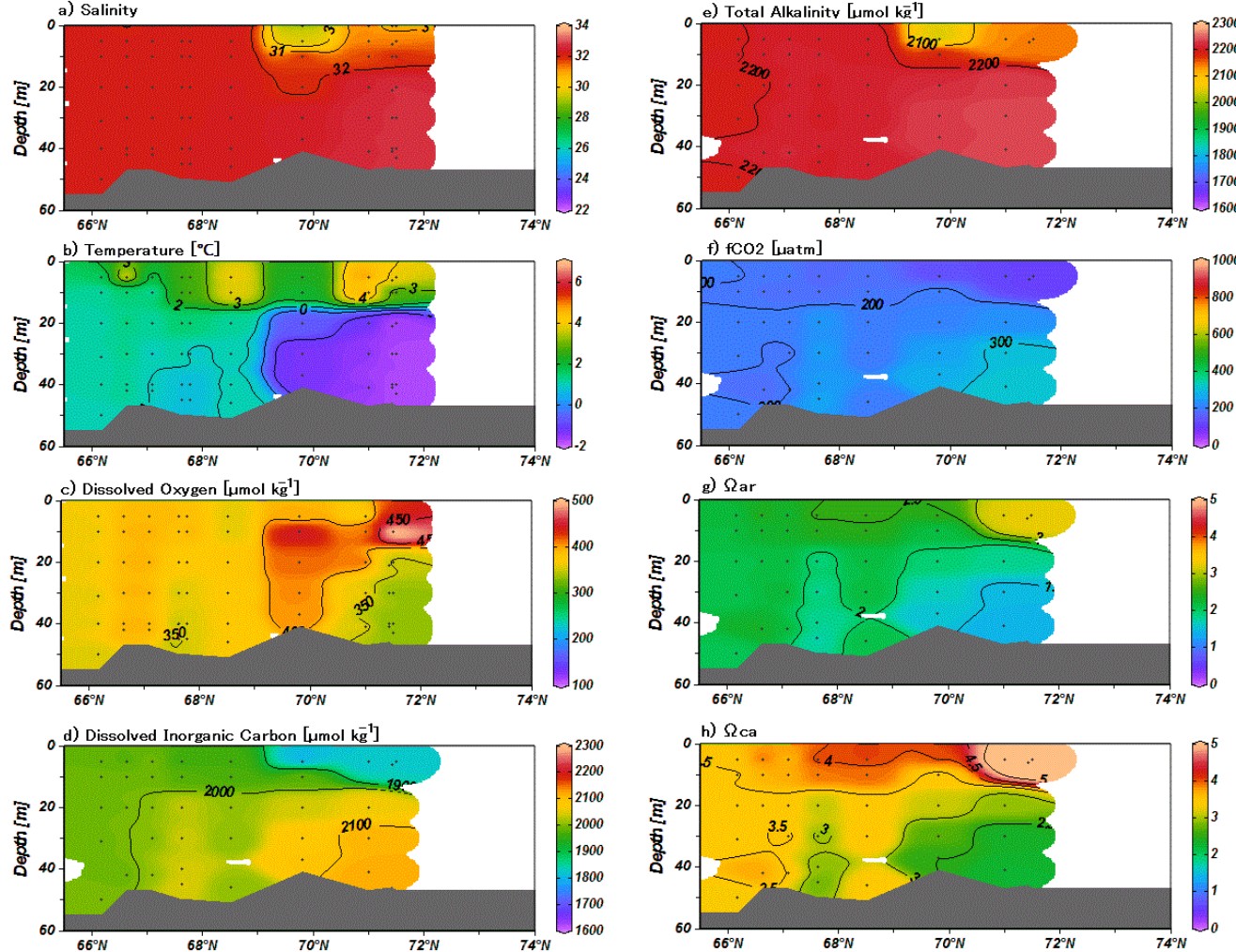

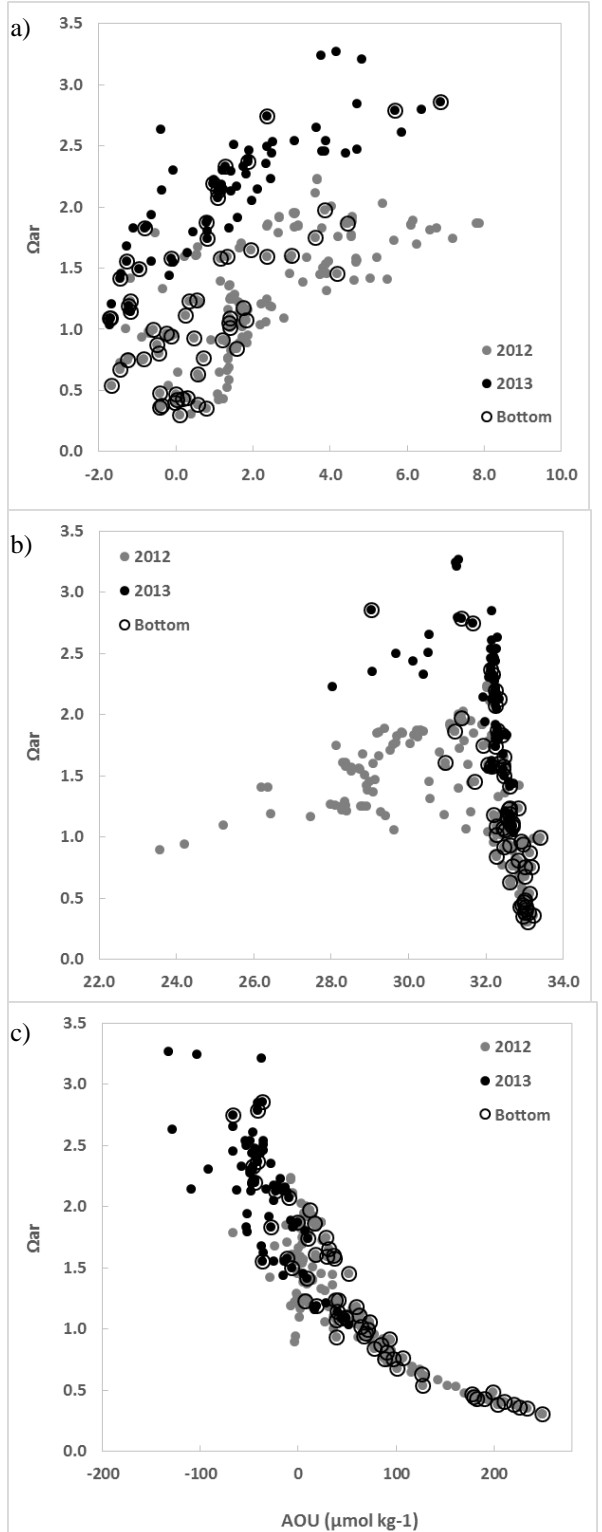

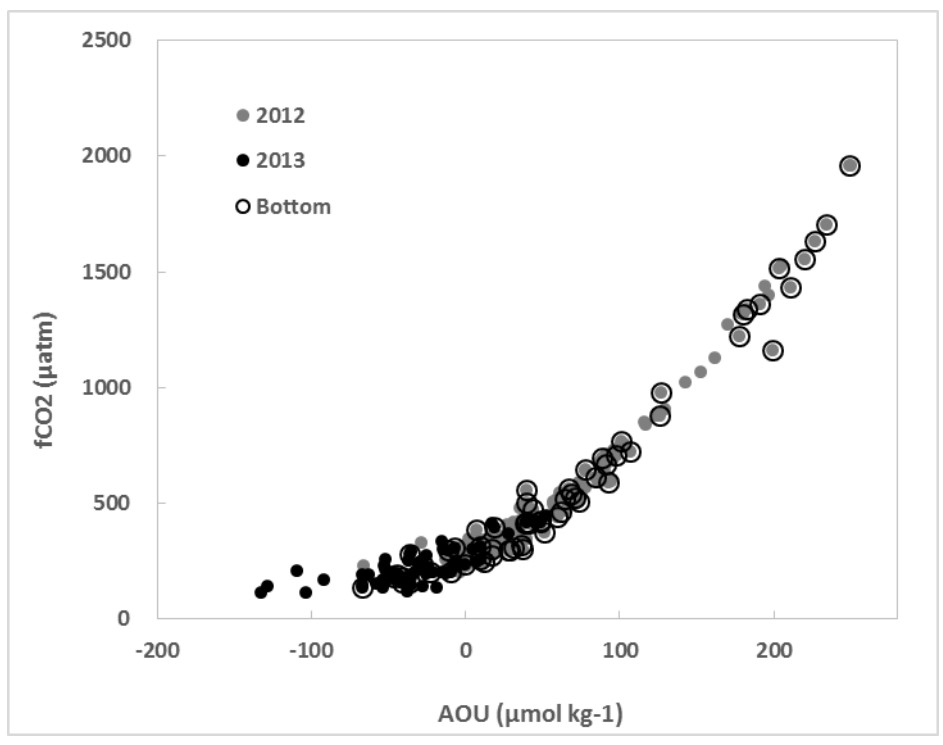

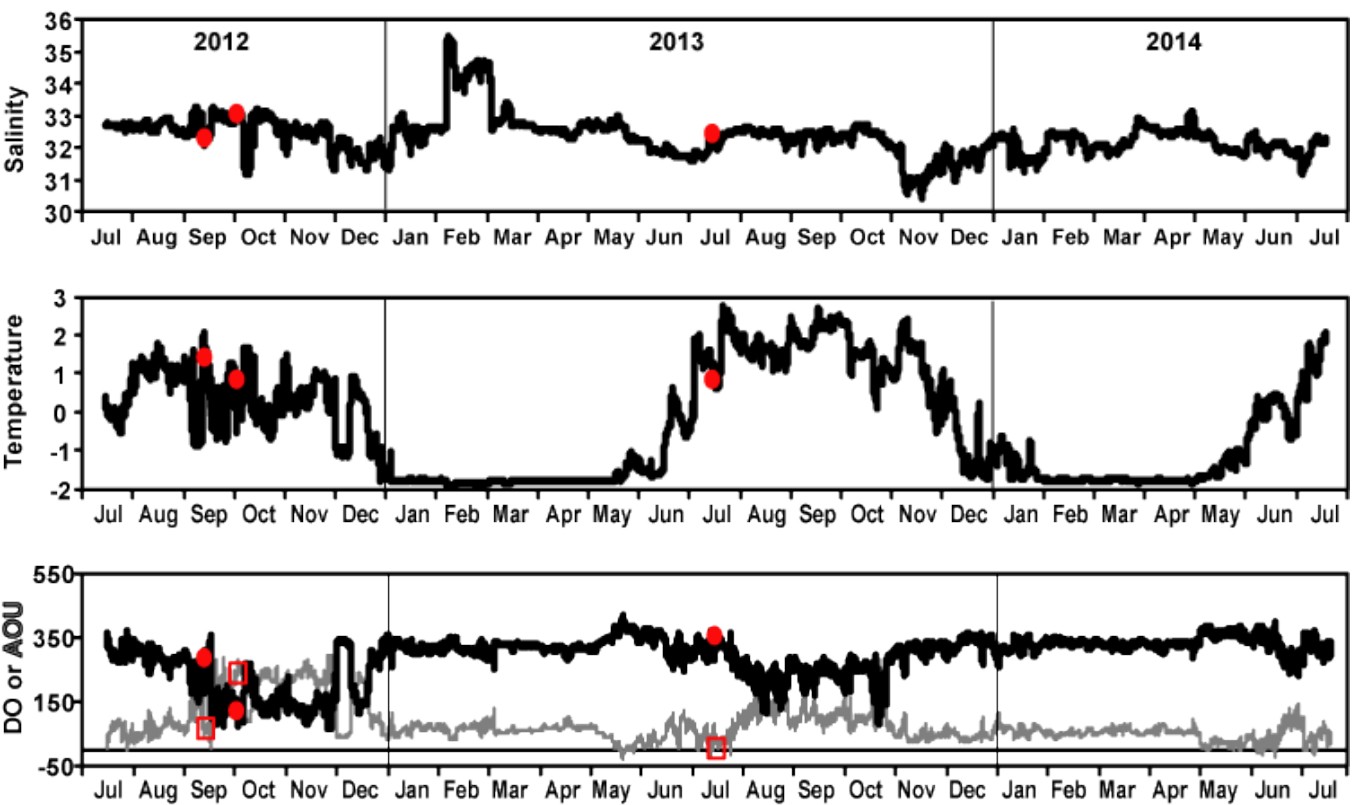

a)

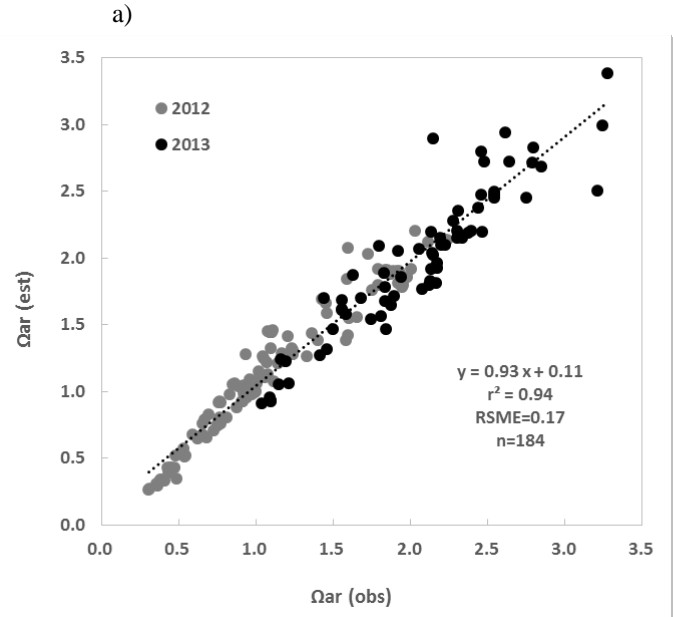

b)

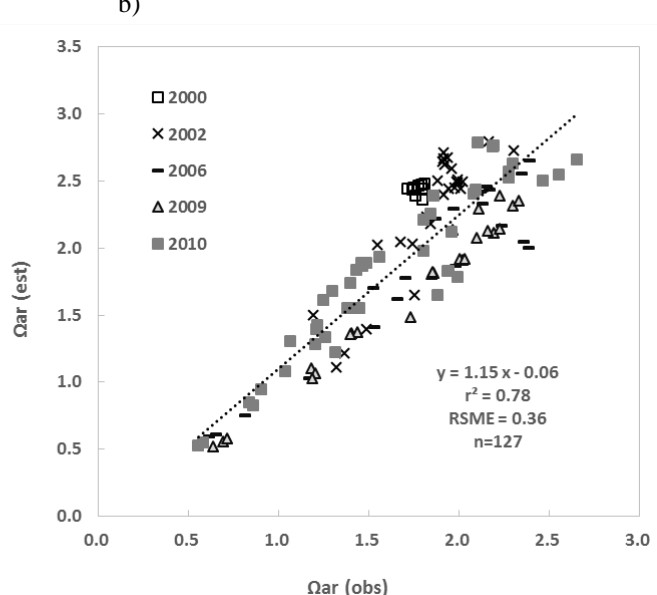

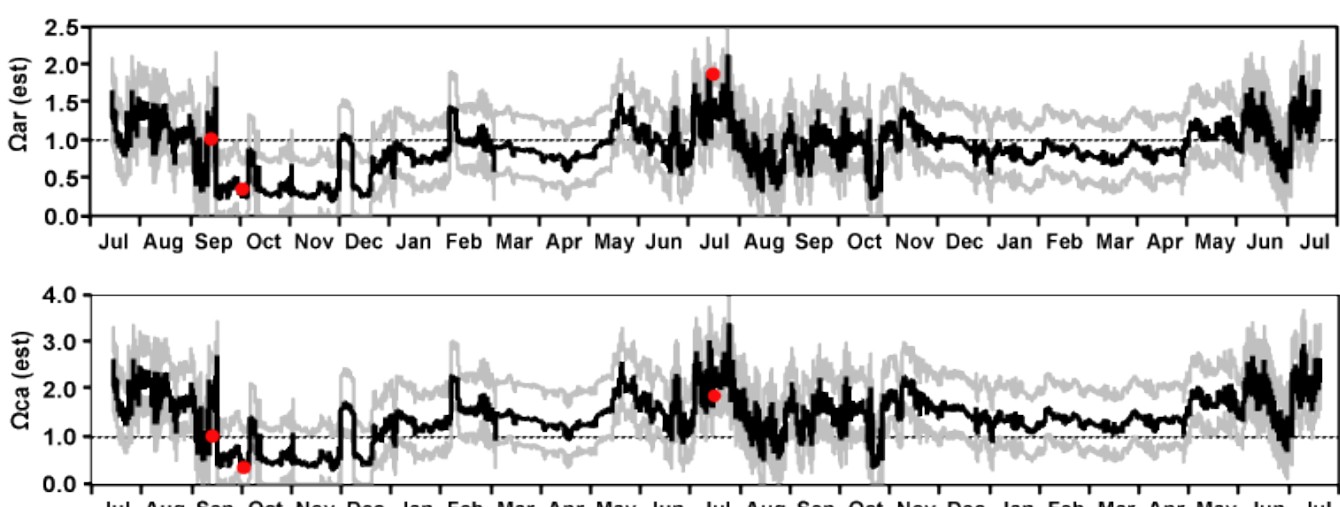

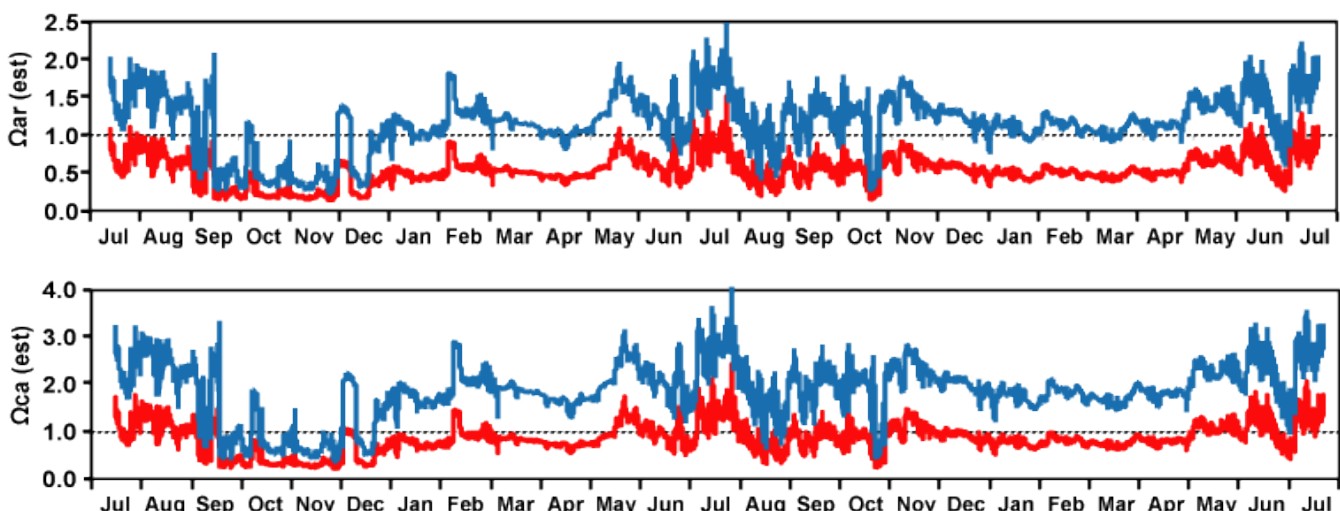