# Peer review of "Seasonal variation of CaCO3 saturation state in bottom water of a biological hotspot in the Chukchi Sea, Arctic Ocean"

_Biogeosciences, 2016_

## Referee Comment (RC2)

**Review of Yamamoto-Kawai et al Biogeosciences Discussion doi:10.5194/bg-2016-74**

The manuscript describes an important issue regarding challenges to monitor ocean acidification in a highly dynamic area with large seasonal and interannual variability. The described method uses measurements and data using shipboard sampling and conventional (relatively, although the TA method using spectrophotometry is not entirely established) methods during 2 weeks each in 2012 and 2013 (both in summer time) to create algorithms. These algorithms are developed as to provide a method to achieve data on the carbonate system and CaCO3 saturation using proxies from sensors attached to moorings in the Chukchi Sea/(Beaufort shelf). The method has been relatively successful in a few other areas (Alin et al..). However, there are several precautions to consider for the Arctic shelves which probably show larger seasonal and interannual variability, in a way also shown in the large difference in the oxygen concentration levels in the bottom water between the two years as presented here. The ship-board data is used in regression analysis using proxies to provide algorithms to determine the variability in the carbonate system at other times of the same year and even for other years. This requires knowledge on the study area and its major drivers to discern the appropriate proxies to describe the carbonate system and CaCO3 saturation. It also require indepth analysis of the uncertainties and the cumulative errors in the methodology. Unfortunately, this type of error analysis is largely lacking and also it is not entirely clear how the proxies were chosen. The authors also "press on" to use the algorithms, not only for several years where there is very little ship measurements to compare with, but also make farfetched and highly speculative conclusions on the causes for decreased saturation in the area. The manuscript also seems to be written in haste (check spelling of co-authors and references) and needs a thorough language check. There are also many parts with redundant or repeating text, sometimes the same sentences appear a few sentences apart, also, the word "hotspot" is used 14 times in the text and most of them could be removed. Part of the redundancy could be arranged easily if the authors include a section describing the study area, including the physical oceanography, explain causes for this area to be a hotspot for biology as well as the ice conditions for the two shipboard measurement years. This would also facilitate the interpretation of the differences between the two years later in the results section. The authors also refer to trends of wide spread and increasing undersaturation but show clearly that the undersaturation they found in 2012 was not present in 2013! Moreover, the large uncertainty in the analysis (shown as range of RMSE as shown in Figure 6), clearly shows that it is not possible to state prolonged undersaturation as the title does. This is biased statement, the positive RMSE do not show undersaturation, neither to ship board data for 2013!. Unfortunately, I cannot recommend this manuscript for publication as it is now and there are substantial revisions and clarifications to be made to make it possible for publication. I suggest that the authors rewrite the manuscript focusing on the methodology and describe it in a "good-honest" manner, including sensor and instrument information (resolution, response time, brand, latest calibration). The regression analysis is lacking information and includes large errors (RMSE) which are referred as "well correlated". One example of limitations in the manuscript is that the algorithm is not validated using independent data on DIC and TA, and the authors use it on several years prior to the data used for the development of algorithms where sensor data was collected. However, during this >10 year of time, was the sensors the same? How frequently were they calibrated? The year that is selected needs to be possible to validate with independent data not only the advantages. Moreover, the limitations to use this method as well as a true and honest error analysis rather than try to push observational and predictive powers that is clearly not within reach for this method in this highly dynamic area, (part of the reason why this is a hotspot).

Major issues summary:

- 1) Methodology lacks important information on analytical methods and sensor information. Show also DIC and TA data as well as AT vs S relationship. Also, the manuscript would benefit from showing  $fCO_2$  (µatm) in relation to DO since that could provide information on the causes of the difference between the two years.
- 2) The regression analysis does not fulfill a proper error progression and is not open about the limitations and the true predictive capacity. The year 2012 is described as an anomalous year. How can it then be used for extrapolation in both time and space in the regression analysis and further in the text? The analysis should also use published data for validation and check the spatial validity of the algorithms. Probably they are very limited and cannot be extrapolated to any other region. RMSE are large for DIC and TA, what consequences do that have for the calculated  $\Omega$ ? Explain.
- 3) The uncertainties in the calculations of the anthropogenic  $CO_2$  impact is not clearly described and emphasized in the abstract without a solid ground.

**More specifically:**

**Title:** Since 2013 did not show undersaturation, also the large range of RMSE also show "prolonged" oversaturation! The title is biased towards showing undersaturation and overstates the "negative" results and does not mirror the large uncertainty. Suggest to add "possible" or focus in the 2012 event with largely undersaturated waters due to biological processes (likely).

**Abstract:**

Include range of saturation instead of only mentioning undersaturation

Row 16-: Highly speculative statement that it is anthropogenic  $CO_2$  uptake that drives the duration of  $CaCO_3$  saturation in the bottom waters. See also later comment. The authors mention the limitations of the method they used in the actual chapter which is not the case in the abstract. Rephrase this part to give possible rates of change per decade at a moderate scenario. RC8.5 is considered unrealistic.

**Introduction:**

Row 16 to 17: This is not correct. Just because it is a shallow shelf it does not mean that the Chukchi Sea has large content of anthropogenic  $CO_2$ . One of the largest content of anthropogenic  $CO_2$  is found in the North Atlantic, which is several thousand meters deep. Use a solid reference if that is the case, if not, delete sentence.

**2. Observation and analysis**

Generally, lack of detailed information on methodology

Row: 5 to 7. Two different methods are used to determine TA, spectrophotomery and potentiometric titration. How did these two methods differ? Was this assessed? The quality of

the TA data plays a large role for the determination of the  $CaCO_3$  saturation. Is there any comparison performed between these two methods? Or was another parameter measured for example pH, as to perform internal consistency check on the quality of the TA and DIC? This information would greatly support some of the findings later in the manuscript and also strengthen the algorithm development.

Row 5: How was dissolved oxygen measured on the ship? What was the method for nutrients? If they were sensor data should also include information on what sensors and how and when they were calibrated. Perhaps the difference on DO in bottom water (Figure 2) between 2012 and 2013 only due to sensor differences? That should be explained.

Row 8 to 10: The precision for DIC is quite high, was there a reason for that? How much will that error result in calculated  $\Omega$ ?

Row 12: Why was Lueker constants chosen? It is preferable if the authors would present results from several other determinations of constants, estimate the mean for all and deviation from the mean for each constant as to assess the range in uncertainty/error by using different set of constants.

Row 17: Describe the sensors (brand), resolution and how the accuracy was established. Row 17: Was the sensor measuring chlorophyll- a or fluorescence? What sensor was used? Was it calibrated?

**Results and Discussion**

**3.1 Ship based observations**

Since the Figure does not refer to any of the sampling locations such as (Pt Barrow or Hope valley) it is difficult to follow. Please add this information in the figure referred to in the text.

The authors should also show the TA and DIC not only  $CaCO_3$  saturation. The discussing that follows on the differences between the two years would greatly benefit from analysis of the TA and DIC data.

Row 18 to 20: Section is repetitive and is almost the same as stated in introduction, redundant.

Row 24: Interesting that undersaturation was found in the Bering Strait. This was not found in 7-years earlier (water column data summer 2005 by Chierici and Fransson (2009). Could that support the statement that undersaturation has progressed in the area? Interesting comparison to add. They also discuss different constants and the result in  $\Omega$  (see previous comment).

Row 32-end of section: The authors explain the difference between the two years of shipboard measurements to be caused by differences in organic matter accumulation. Was there evidence for larger primary production in 2012 than in 2013? Perhaps sediment trap data? Or satellite data showing differences in primary production? Later in the manuscript the authors refer to Grebmeier et al 2015 for chlorophyll a data. This could be developed further.

**3.2: Mooring observations**

Please point out where the mooring was located in Figure 1.

Row 3: Display that sensor data agreed well, how well? Show some numbers?

Row 6: what does the author mean by" T, S, DO showed larger and high frequent variability"?

The changes in water masses limit the possibility to use algorithms to estimate aragonite saturation in this area, since it is highly variable. Should add information on the limitations of the spatial extent for the algorithms.

Row 11-12: What about the ice conditions between the two years? Could variability in sea ice formation/melting cause these changes? Did they differ between the years?

**3.3 Regression analysis**

Generally: This chapter is not open on its limitations, and does not give a ground-based insight into the development of algorithms or the predictive power of the proxies. This requires a much more thorough and detailed and unbiased description of the methodology that is used here and its limitations. For example: What is the cumulative error in the regression analysis including all error (analytical methods, CO2SYS calculation, difference between observed and estimated). The Figures where this data is used should all include error bars.

Row 15: DIC is also influenced by mixing of watermasses, not only TA.

Row 17: What about calcification by primary producers in the area? That could also cause changes in the TA.

Row 19: the authors should include the TA -S figure. This will add information on the scatter in the data.

Why was the chl a not used? Chl a is commonly used as proxy for primary production. AOU could be a proxy for respiration. Check also with fCO2 which generally correlates well with DO. If they do not correlate that could help in explaining other causes for differences. Explain the choices better.

Row 20: First the authors mention that the best predictions were obtained using all three proxies, then they end up using fewer and fewer, this is understandable since it is better to be able to estimate the whole carbonate system as simple as possible. But it is not convincing and do not explain the selection of proxies. Most published algorithm development use a stepwise method to go forward with the proxy selection.

Moreover, the RMSE for TA is about 14  $\mu$ mol/kg, which is quite substantial. What is the consequence for the calculated  $\Omega$ ? Need also to consider the RMSE i DIC which is even larger (24  $\mu$ mol/kg). The authors need to address what that means for the calculated  $\Omega$  values.

Row 30: the RMSE for aragonite and calcite are quite large and cannot be neglected. The authors should perform a proper error analysis and estimate the cumulative error and discuss the error in context of the seasonal and interannual differences they found. This is crucial for the discussions after on trends and future projections.

Row 29: What is the cause for the large difference in the surface waters? Explain What is the RMSE for the fits mentioned here?

Row 32. It is repeatedly displayed in the results between the two years that this is a highly variable and dynamic area. This also means that the algorithms are likely to be highly uncertain and result in high bias for other times of the year and also between years. Thus, using the algorithms for 2000, 2002, 2006, 2009, and 2010 is highly speculative. The authors states that this is a validation of the method but to be a validation the authors need independent data on shipboard DIC and TA to compare with the estimated DIC and TA from the algorithms. This is a great weakness in the method and I think the authors should remove this analysis since it is highly speculative. Especially since the authors later on uses results from this analysis to make projections on future saturation states. There are likely other published DIC and TA data in this region which could be used for a proper validation. This area is one of the most studied in the Arctic Ocean with regard to the carbonate system.

Moreover, using sensor data from more than 10 years must give details on sensor types, were they the same for all those years? Calibrations? Precision? Drift?

Page 7: Row 1-4: the authors mention that the estimated and observed omega values correlate well, RMSE is 0.36 and 0.57!! this give a very large relative standard deviation (CV%) compared to the mean  $\Omega$ , and is not what should be referred as "well correlated". Again, needs a proper error analysis and put into context of the seasonal and interannual variability in the omega values. Be more objective.

**3.4.**

Row 13-14: repeat biological hotspot again.

Page 8: Row10: If 2012 was such an unusual year (or autumn), how can the algorithm that is developed on this data give reliable estimates for other years (and other areas)? Sometimes 10 years back in time? Not solid and again shows that the data from this study area cannot be extrapolated to other times of the year nor to other years.

**3.5 Anthropogenic impact**

The used method assumes that the rate of uptake in the bottom water is the same as in the surface water uptake of anthropogenic  $CO_2$ . That should be added to the limitations of this method. The method also assumes that ocean mixing and all other processes have remained the same, which is partly commented by the authors regarding biological processes. It could be worthwhile using reported values of decadal uptake rates from for example Tanhua et al., 2007 Anthropogenic  $CO_2$  in the Arctic Ocean) and use that rate of change to estimate the impact of anthropogenic uptake. That is more robust and also shows the rate of change at depth, which is likely not the same as in the surface waters. Regardless the authors should include more on the assumptions and limitations on the resulting omega values.

Also, after the last Paris (COP) meeting it was agreed that the RCP8.5 scenario is unrealistic and I suggest that the authors include a moderate scenario in addition.

Also, now it is difficult to follow the actual change in omega, it would be better to suggest a rate of change /decade. How much would omega change due to anthropogenic  $CO_2$  per decade?

**4. Summary and conclusions**

Rewrite summary with regard to previous comments. Biological hotspot is used many times. Reference spelling! Kroeder!?

**Figures:**

Figure 1: Please show the main study sites on the map Pt Barrow and Hope Valley, Bering Strait.

Figure 2: Figure 2c: add abbreviation DO which is used in the text and unit for DO. Add plots showing DIC, TA and  $fCO_2$ . Include units. Moreover, there is no undersaturation in 2013 which is not the impression given in the text. Please, clarify in the text.

Figure 3: Describe what a negative and positive AOU refers to.

Figure 4: Difficult to see the bias between shipboard and sensor data in the plot, suggest to add a Table showing the differences. The Table should also include the number of samples/data points that were compared (N).

Figure 5a: Show the data from 2012 and 2013 i different symbols or colors. Figure 5b (x-axis), is the  $\Omega$ obs based on 2012 and 2013 data? Clarify. Are the displayed data mean values for all years between 2000-2010 or separate years? If separate years, show the different years using different symbols for improved understanding of the bias to the observed for the different years.

Figure 6: The errors and biases are large as is shown in the RMSE (grey lines). This error "only" includes the error from reconstruction and do not include the cumulative error from the  $\Omega$  calculation from DIC and TA. The figure text should include the full regression incl coefficient of correlation.

Figure 7: Perhaps combine Figure 6 and Figure 7 to show present, pre-ind and a future 50 years.

Figure 8: Redundant picture. Also, Figure caption text is wrong, should probably be "collected".

---

## Referee Comment (RC1) · L.ÂăW. Cooper (Referee) · 21 Apr 2016

Review of Yamamoto-Kawai et al. Biogeosciences Discuss., doi:10.5194/bg-2016-74, 2016

This is a short paper that uses shipboard observations of aragonite undersaturation (via alkalinity and DIC measurements) in the Chukchi Sea in 2012 and 2013 to describe regions of the Chukchi Sea where undersaturation was present during summer and autumn measurements. The authors also use data on oxygen utilization, temperature and salinity from those two cruises to see how well those measurements can predict the estimated  $\Omega$  value of carbonate saturation. They then use these empirical correlations to estimate the degree of undersaturation of calcite and aragonite at a moored location where dissolved oxygen, salinity, temperature, and chlorophyll measurements were also made over a two-year period. They find that in this moored location with high productivity that undersaturation is likely widespread and lengthy over the course of the year, although the actual impacts of calcium carbonate dissolution on benthic communities in this area do not seem obvious, based upon sampling of clam communities on the bottom. Obviously the organisms that inhabit this area and take advantage of the abundant food supplies have resiliency with respect to ocean acidification.

The observational portions of the study showing undersaturation during the summer and fall sampling periods are not surprising given other published work in the Chukchi Sea, but still add to our understanding of widespread undersaturation impacts. I have more misgivings about the extension of using the moored data and apparent correlations developed between AOU and ocean acidification to estimate undersaturation of calcite minerals over the course of the year. While I am not surprised that undersaturation is probably common due to mineralization and high productivity, the conclusions are based upon the assumption that oxygen utilization continues at fairly constant rates over the winter, and I think the small published set of sediment oxygen utilization measurements available from arctic shelves does not strongly support this assumption. Only one study (Devol et al. 1997) is cited to support this assumption, and it sampled in the winter in unproductive waters much different from the moored site. The moored data used (Nishino et al. 2016) also had to be managed-corrections undertaken for AOU data that were corrected because of apparent issues with the data that are mentioned in Nishino et al. 2016. Finally, the use of this correlation method for estimating calcium carbonate dissolution potential was initially demonstrated in California and Oregon, so it really hasn't been confirmed to work in the Arctic where there are much more extreme seasonal changes in biological activity. The authors defend their approach by stating that their shipboard sampling bracketed both high productivity in July and high oxygen utilization in October although my examination of the Nishino et al 2016 results suggest that sampling in July may have missed the highest primary
I don't think this is necessarily a flawed paper because the available evidence suggests that widespread undersaturation with respect to carbonate minerals on productive arctic shelves is probably correct, but I don't think the evidence provided here is strongly convincing either. The manuscript is also unevenly written, and would benefit from efforts of a native English language editor. A number of mistakes in spelling, in the references, and even in the spelling of the author names suggest a hasty assembly of the manuscript. I have provided some editing suggestions below and posed a few additional questions and concerns, but this is not a comprehensive editing effort.

Page 1: Line 3. I think Dr. Nishino's name is misspelled.

Page 1: Line 26. Change "to affect" to "which affects"

Page 2: Line 3. Salisbury et al. reference misspelled.

Page 2: Line 7. Change "Nutrients....is carried" to "Nutrients....are carried"

Page 2: Line 8. Change "making the sea to have very high primary productivity" to "promoting very high primary productivity"

Page 2: Line 9. Add the article "A" before "proportion"

Page 2: Line 14. Change spices to species

Page 2: Line 22. Lower case 3 needed for calcium carbonate molecular symbol.

Page 2: Line 24. Change "difficulties in" to "the lack of"

Page 2: Line 25. I don't follow why the reference to Talmange and Gobler, 2009 needs to be made here. This reference has already been made (prior page, Line 29) to document larval stage vulnerability, although that reference is about non-polar invertebrates. The statement and reference repeated here is redundant

Page 3: line 2. Change was to were

BGD
Page 3: line 15. "the" before maintenance not necessary

Page 3: line 27. Delete "that" and change "is" to "as"

Page 3: Line 29. Insert a "the" before "two visits"

Page 4: Line 10. Change "kept at near" to "remained at a near"

Page 5: Line 23. Change kept to remained

Page 5: Line 31. Change captured to sampled

Page 5: Line 27-30. Most of the published data for sediment oxygen utilization rates for the northern Bering and Chukchi seas indicates that there is significant seasonal variation and it is lower in the late winter prior to initiation of the sea ice edge bloom. I think the Devol et al. paper is dubious to cite here because the winter sampling was done in nutrient-poor, near-shore waters that do not have high AOU at any time of year.

Page 6: Line 26. Change "in" to "to a"

Page 7: Line 15. Remove "of" The sentence would also read better if it starts with the article "the"

- Page 7: Line 25. Suggest should be suggests.
- Page 7: Line 28. Devol reference should be 1997, not 1996.
- Page 7: Line 30 persisted should be persistent.
- Page 8. Line 16. This really isn't a complete sentence.
- Page 8. Line 31. Change "process" to "processes" and "is" to "are"
- Page 9. Line 1. Change is to are
- Page 9. Line 2. Add "the" between that and primary production.
- Page 9. Line 3. There is a Grebmeier, 2012 reference in the literature cited, but not a
Grebmeier et al. 2012.

Page 9. Line 6-7. The sentence is not grammatically correct and I am not sure what the authors are trying to say.

Page 9. Line 9. Change "even with half productivity than today" to "even with half the productivity occurring today"

Page 9. Line 15. Change "it is indicated" to "it suggests"

Page 9. Line 16. Change "occupation" to "the proportion of"

Page 9. Line 17. Change "occupies" to "increases to"

Page 9. Line 18. Change These to This and indicate to indicates; add the article "a" has and significant.

Page 9. Line 22. I suggest changing Horizontal to Spatial

Page 9. Line 25. Change "to" to "from"

Page 9. Line 26. The mooring observations are presented in Nishino et al. 2016, so I think it is more accurate to state that the authors used the data from Nishino et al. 2016 to estimate calcium carbonate undersaturation.

Page 9. Lines 27-29. The Nishino et al. 2016 paper appears to show that the maximum chlorophyll a bloom can occur prior to July, so the early summer sampling may not have successfully sampled the most productive period.

Page 9. Line 33. Change "Occupation of calcite" to "The period of calcite"

Page 10, Line 2. Insert "subject to" between "been" and "aragonite"

Page 10. Line 5. Change two-hold to two-fold; change "occupation" to "the period of"

Page 10. Line 6. Change "year-long occupation under highly stratified condition. Occupation..." to "year-round periods under highly stratified conditions. Periods of
Page 10. Line 8. I suggest changing "surely" to "clearly". It is less colloquial and more specific

Page 10. Line 10. Kroeker's name is misspelled.

"

Page 10. Line 12. Change "may be conflicting the fact" to "is not consistent with the fact"

Page 10. Line 29. Since there was no formal presentation of oxygen isotope data, I don't think an acknowledgement is necessary.

Page 12. Line 17. Global Change Boil should be Global Change Biol

Page 13, line 25. Raven reference is not in alphabetical order.

Figure 1. The arrows identifying the mooring sites are not clear.

Figure 8 caption. Corrected should be collected. Also trawl is misspelled.

---

## Author Comment (AC1) · 18 Jun 2016

**First of all, we would like to thank Professor Lee Cooper for a thorough review of this paper. We have addressed all comments and suggestions. Please find our responses below**

I have more misgivings about the extension of using the moored data and apparent correlations developed between AOU and ocean acidification to estimate undersaturation of calcite minerals over the course of the year. While I am not surprised that undersaturation is probably common due to mineralization and high productivity, the conclusions are based upon the assumption that oxygen utilization continues at fairly constant rates over the winter, and I think the small published set of sediment oxygen utilization measurements available from arctic shelves does not strongly support this assumption. Only one study (Devol et al. 1997) is cited to support this assumption, and it sampled in the winter in unproductive waters much different from the moored site.

--We agree that we need to discuss more about the winter AOU and its correlation to our reconstruction of CaCO3 saturation state. A relatively constant positive AOU (~50 µmol kg1) was observed over the winter for two years. As mentioned in the text, same level of AOU was also found in the hypersaline water that is formed in contact with atmosphere. This suggests that positive AOU in winter bottom water is not due to insufficient gas exchange but oxygen consumption. Although there is no year-round observation of sediment oxygen uptake in southern Chukchi Sea, it is known that oxygen uptake rate has a seasonal variation and is low in winter prior to initiation of biological production in spring. Previous studies in other Arctic waters have observed that sediment oxygen uptake rate in winter is not zero but is about half of that in summer in coastal area north of Pt. Barrow, Alaska (Devol et al., 1997), in Young Sound in Northeast Greenland (Rysgaard et al., 1998) and in Resolute Bay in Canadian high Arctic (Welch et al., 1997). Winter AOU observed by moored sensor in our study was about half of autumn AOU in 2012 and 1/4 in 2013. Therefore, we presume that the positive AOU in bottom water during winter can be explained by continued sediment oxygen uptake. This means that correlation between AOU and DIC should hold in winter bottom water. This assumption should be verified by winter observation of carbonate parameters by  $pCO_2$  or pH sensors, automatic water samplers, or winter cruise observation in the future. These discussions have been included in the revised text as follows:

"Continued sediment oxygen uptake is a possible reason for the positive and constant AOU in bottom water during winter. Previous studies in shallow Arctic seas have found that sediment oxygen uptake rate is regulated by the availability of organic matter and macrofaunal biomass (Grebmeier and McRoy, 1989; Rysgaard et a l., 1998; Grant et al., 2002; Clough et al., 2005). Accordingly, oxygen uptake rate has a seasonal variation and is low in winter prior to initiation of biological production in spring (Cooper et al., 2002; Grant et al., 2002;). Nevertheless, winter sediment oxygen uptake rate is not zero but is about half of that in summer in coastal area north of Pt. Barrow, Alaska (Devol et al., 1997), in Young Sound in Northeast Greenland (Rysgaard et al., 1998) and in Resolute Bay in Canadian high Arctic (Welch et al., 1997). Winter AOU observed in our study was about half and 1/4 of autumn AOU in 2012 and 2013, respectively (Figure 4). This does not contradict observed seasonality in sediment oxygen uptake.

"Estimated low  $\Omega$  in winter is likely due to continued oxygen uptake by benthic organisms during winter as suggested by positive AOU. Note that we applied the regression equation obtained from summer/autumn cruise observations to estimate winter  $\Omega$ . We believe this is acceptable as because remineralization of organic matter should change AOU and DIC at a similar rate regardless. This assumption should be verified by winter observation of carbonate parameters by ship-based sampling, pCO2 or pH sensors, or automatic water samplers in the future."

The moored data used (Nishino et al. 2016) also had to be managed—corrections undertaken for AOU data that were corrected because of apparent issues with the data that are mentioned in Nishino et al. 2016.

--We have carefully checked data and calculations and found that there was a mistake in unit conversion (from saturation % to µmol/kg) in Nishino et al. (2016) for 2013 mooring data. This was the cause of the large difference of 69 µmol/kg between bottle and sensor measurements mentioned in their paper. With correct unit conversion, the difference was only 4 µmol/kg. In our original manuscript, we did not use calculations by Nishino et al. (2016) and used data correctly converted from original sensor output. Therefore, this mistake does not affect our results. This issue is now mentioned in the text to not cause same concern to readers: "Note that an offset of 69 µmol/kg found in sensor DO data for 2013 mooring in Nishino et al. (2016) was due to an artificial error in conversion of original sensor output to µmol/kg. With correct conversion, difference between sensor bottle DO data was only 4 umol/kg. Accordingly, we did not apply any correction to DO sensor data in the present paper."

Finally, the use of this correlation method for estimating calcium carbonate dissolution potential was initially demonstrated in California and Oregon, so it really hasn't been confirmed to work in the Arctic where there are much more extreme seasonal changes

**in biological activity.**

- --Our study is the first attempt to use this method to highly productive Arctic shelf sea. We agree that this should be confirmed in the future by using direct observations of carbonate parameters throughout the year. We have noted this in the revised text to read: "We should note this study is the first attempt to reconstruct seasonal variation of  $\Omega$  using a method that has not been confirmed to work in Arctic shelf seas where seasonal changes in biological activity are extremely large. Direct observation of carbonate parameters in winter by using sensors or water sampler is desired to confirm our results.
- The authors defend their approach by stating that their shipboard sampling bracketed both high productivity in July and high oxygen utilization in October although my examination of the Nishino et al 2016 results suggest that sampling in July may have missed the highest primary productivity.
- --We agree that the maximum chlorophyll a was observed in May/June. However, our two shipboard samplings were made in two different period with high and low DO conditions. To be more accurate, text has been modified from "ship-based observations captured both higher and lower ends of seasonal variation in DO" to "ship-based observations captured both higher and lower parts of seasonal variation in DO", from "our ship-based observations in autumn 2012 and summer 2013 have captured the lowest and the highest  $\Omega$  periods, respectively." To "our ship-based observations in autumn 2012 and summer 2013 have captured low and high  $\Omega$  periods, respectively."
- I don't think this is necessarily a flawed paper because the available evidence suggests that widespread undersaturation with respect to carbonate minerals on productive arctic shelves is probably correct, but I don't think the evidence provided here is strongly convincing either.
- --We have revised the manuscript to describe results in an honest manner. Sentences have been changed to be more exact and fair, for example, "bottom water was kept at aragonite undersaturation for most of the winter" was changed to " $\Omega$  in bottom water was kept low during winter" and "intermittent undersaturation was found" was changed to "intermittent undersaturation was suggested".
- The title was also change from "prolonged undersaturation..." to "Seasonal variation of  $CaCO_3$  saturation state in bottom water of a biological hotspot in the Chukchi Sea, Arctic Ocean".
- We hope the revised manuscript will meet the requirements for publication.

- The manuscript is also unevenly written, and would benefit from efforts of a native English language editor. A number of mistakes in spelling, in the references, and even in the spelling of the author names suggest a hasty assembly of the manuscript. I have provided some editing suggestions below and posed a few additional questions and concerns, but this is not a comprehensive editing effort.
- --We are sorry that we have submitted the manuscript with many typographical errors and really appreciate your kind editing. We will ask an English Language Service to edit our revised manuscript.
- Page 2: Line 25. I don't follow why the reference to Talmange and Gobler, 2009 needs to be made here. This reference has already been made (prior page, Line 29) to document larval stage vulnerability, although that reference is about non-polar invertebrates. The statement and reference repeated here is redundant

--The reference has been removed. The sentence has been modified to read "Because many benthic organisms have planktonic larval stages, timing and duration of  $CaCO_3$ can be critical for their growth and populations".

- Page 5: Line 27-30. Most of the published data for sediment oxygen utilization rates for the northern Bering and Chukchi seas indicates that there is significant seasonal variation and it is lower in the late winter prior to initiation of the sea ice edge bloom. I think the Devol et al. paper is dubious to cite here because the winter sampling was done in nutrient-poor, near-shore waters that do not have high AOU at any time of year.
- --See response to the first comment above.

Page 8. Line 16. This really isn't a complete sentence.

--The text has been changed as follows:

"In order to quantify the effect of anthropogenic CO2 on our 2-year time series of  $\Omega$ , we have estimated  $\Omega$  for two cases: 1) preindustrial period case with pCO2=280ppm, and 2) future case with pCO2=650 ppm. Following previous studies (Gruber et al., 1996; Sabine et al., 1999; Yamamoto-Kawai et al., 2013; 2015), DIC concentration observed in year to obs is expressed as: DICt-obs = DICEQt-0 + ( $\Delta$ diseq +  $\Delta$ bio), where....".

Page 9. Line 6-7. The sentence is not grammatically correct and I am not sure what the authors are trying to say.

--The sentence has been deleted.

**Page 9. Line 25. Change "to" to "from"**

- --We could not find "to" in this line and are not sure where the reviewer found this error. This will be corrected when we have the professional English editing.
- Page 9. Lines 27-29. The Nishino et al. 2016 paper appears to show that the maximum chlorophyll a bloom can occur prior to July, so the early summer sampling may not have successfully sampled the most productive period.
- --We agree that chlorophyll a was highest in June, though it is still high in July. The text has been changed from "under photosynthesis in early summer 2013" to "under an influence of photosynthesis".

**All of following have been changed as suggested.**

Page 1: Line 3. I think Dr. Nishino's name is misspelled.

Page 1: Line 26. Change "to affect" to "which affects"

Page 2: Line 3. Salisbury et al. reference misspelled.

Page 2: Line 7. Change "Nutrients. . .. is carried" to "Nutrients. . .. are carried"

Page 2: Line 8. Change "making the sea to have very high primary productivity" to "promoting very high primary productivity"

Page 2: Line 9. Add the article "A" before "proportion"

Page 2: Line 14. Change spices to species

Page 2: Line 22. Lower case 3 needed for calcium carbonate molecular symbol.

Page 2: Line 24. Change "difficulties in" to "the lack of"

Page 3: line 2. Change was to were

Page 3: line 15. "the" before maintenance not necessary

Page 3: line 27. Delete "that" and change "is" to "as"

Page 3: Line 29. Insert a "the" before "two visits"

Page 4: Line 10. Change "kept at near" to "remained at a near"

Page 5: Line 23. Change kept to remained

Page 5: Line 31. Change captured to sampled

Page 6: Line 26. Change "in" to "to a"

Page 7: Line 15. Remove "of" The sentence would also read better if it starts with the article "the"

Page 7: Line 25. Suggest should be suggests.

Page 7: Line 28. Devol reference should be 1997, not 1996.

Page 7: Line 30 persisted should be persistent.

Page 8. Line 31. Change "process" to "processes" and "is" to "are"

Page 9. Line 1. Change is to are

Page 9. Line 2. Add "the" between that and primary production.

- Page 9. Line 3. There is a Grebmeier, 2012 reference in the literature cited, but not a Grebmeier et al. 2012.
- Page 9. Line 9. Change "even with half productivity than today" to "even with half the productivity occurring today"

Page 9. Line 15. Change "it is indicated" to "it suggests"

Page 9. Line 16. Change "occupation" to "the proportion of"

Page 9. Line 17. Change "occupies" to "increases to"

Page 9. Line 18. Change These to This and indicate to indicates; add the article "a" has and significant.

Page 9. Line 22. I suggest changing Horizontal to Spatial

- Page 9. Line 26. The mooring observations are presented in Nishino et al. 2016, so I think it is more accurate to state that the authors used the data from Nishino et al. 2016 to estimate calcium carbonate undersaturation.
- Page 9. Line 33. Change "Occupation of calcite" to "The period of calcite"
- Page 10, Line 2. Insert "subject to" between "been" and "aragonite"
- Page 10. Line 5. Change two-hold to two-fold; change "occupation" to "the period of"
- Page 10. Line 6. Change "year-long occupation under highly stratified condition.
- Occupation. . ." to "year-round periods under highly stratified conditions. Periods of"

Page 10. Line 8. I suggest changing "surely" to "clearly". It is less colloquial and more specific

Page 10. Line 10. Kroeker's name is misspelled.

Page 10. Line 12. Change "may be conflicting the fact" to "is not consistent with the fact"

- Page 10. Line 29. Since there was no formal presentation of oxygen isotope data, I don't think an acknowledgement is necessary.
- Page 12. Line 17. Global Change Boil should be Global Change Biol

Page 13, line 25. Raven reference is not in alphabetical order.

Figure 1. The arrows identifying the mooring sites are not clear.

Figure 8 caption. Corrected should be collected. Also trawl is misspelled.

---

## Author Comment (AC2)

- First of all, we would like to thank the reviewer for a review of this paper. We have addressed all comments and suggestions. Please find our responses below
- However, there are several precautions to consider for the Arctic shelves which probably show larger seasonal and interannual variability, in a way also shown in the large difference in the oxygen concentration levels in the bottom water between the two years as presented here. The ship-board data is used in regression analysis using proxies to provide algorithms to determine the variability in the carbonate system at other times of the same year and even for other years. This requires knowledge on the study area and its major drivers to discern the appropriate proxies to describe the carbonate system and CaCO3 saturation. It also require indepth analysis of the uncertainties and the cumulative errors in the methodology.
- Unfortunately, this type of error analysis is largely lacking and also it is not entirely clear how the proxies were chosen. The authors also "press on" to use the algorithms, not only for several years where there is very little ship measurements to compare with, but also make farfetched and highly speculative conclusions on the causes for decreased saturation in the area.
- --Please see below our response to each specific comment.
- The manuscript also seems to be written in haste (check spelling of co-authors and references) and needs a thorough language check. There are also many parts with redundant or repeating text, sometimes the same sentences appear a few sentences apart, also, the word "hotspot" is used 14 times in the text and most of them could be removed. Part of the redundancy could be arranged easily if the authors include a section describing the study area, including the physical oceanography, explain causes for this area to be a hotspot for biology as well as the ice conditions for the two shipboard measurement years. This would also facilitate the interpretation of the differences between the two years later in the results section.
- --We are sorry that we have submitted the manuscript with many typographical errors and redundant text. We will ask an English Language Service to edit our revised manuscript. Many of the word "hotspot" has been removed. We have add a section describing the study area in the revised manuscript. Thank you for your suggestion.
- The authors also refer to trends of wide spread and increasing undersaturation but show clearly that the undersaturation they found in 2012 was not present in 2013!

- --Our ship-based observation in 2013 was performed in high saturation season when waters were under an influence of photosynthesis as suggested by moored sensors of DO and chlorophyll a. We believe that this does not conflict with "increasing undersaturation" with increasing anthropogenic CO2.
- Moreover, the large uncertainty in the analysis (shown as range of RMSE as shown in Figure 6), clearly shows that it is not possible to state prolonged undersaturation as the title does. This is biased statement, the positive RMSE do not show undersaturation, neither to ship board data for 2013!.
- --We agree. The title has been changed to: "Seasonal variation of CaCO3 saturation state in bottom water of a biological hotspot in the Chukchi Sea, Arctic Ocean"
- I suggest that the authors rewrite the manuscript focusing on the methodology and describe it in a "good-honest" manner, including sensor and instrument information (resolution, response time, brand, latest calibration).
- --We have revised the manuscript to describe results in a good honest manner. Sentences have been changed to be more exact and fair, for example, "bottom water was kept at aragonite undersaturation for most of the winter" was changed to " $\Omega$  in bottom water was kept low during winter" and "intermittent undersaturation was found" was changed to "intermittent undersaturation was suggested".

We hope the revised manuscript will meet the requirements for publication. Information of DO sensor has been included (see below).

The regression analysis is lacking information

--Please see below

and includes large errors (RMSE) which are referred as "well correlated". -- The word "well" was removed.

One example of limitations in the manuscript is that the algorithm is not validated using independent data on DIC and TA, and the authors use it on several years prior to the data used for the development of algorithms where sensor data was collected. However, during this >10 year of time, was the sensors the same? How frequently were they calibrated? The year that is selected needs to be possible to validate with independent data not only the advantages. Moreover, the limitations to use this method as well as a true and honest error analysis rather than try to push observational and predictive powers that is clearly not within reach for this method in this highly dynamic area, (part of the reason why this is a hotspot).

--We found that the concern of the reviewer is based on misunderstanding. We have validated the algorithms by comparing independent data of shipboard DIC and TA in 2000, 2002, 2006, 2009 and 2010 with estimated DIC and TA using equations obtained for 2012/2013 data. Comparison showed that the algorism could reproduce  $\Omega$  for multiple cruises even the area has high variability. Therefore we concluded that the algorism is applicable to the study area. We have modified the text to explain this clearly:

"In order to evaluate regression equations obtained from 2012-2013 cruises, we have applied the same equations to independent data from R/V Mirai cruises in the Chukchi Sea in 2000, 2002, 2006, 2009 and 2010..... We have calculated DICest and TAest from T, S and AOU data of these cruises using the equations obtained for 2012-2013. AOU was calculated from bottle DO data. DICest and TAest agreed with independent shipboard observations of DIC and TA with  $r^2=0.96$  and 0.83, respectively.  $\Omega$  calculated from DICest and TAest ( $\Omega$ est) was..."

**Major issues summary:**

- 1) Methodology lacks important information on analytical methods and sensor information.
  - --See our response above.

Show also DIC and TA data as well as AT vs S relationship. Also, the manuscript would benefit from showing fCO2 (µatm) in relation to DO since that could provide information on the causes of the difference between the two years.

--Figures suggested will be added to the revised manuscript. fCO2 and AOU correlated well as described below. Please see figures of S-TA and fCO2-AOU submitted with this reply.

2) The regression analysis does not fulfill a proper error progression and is not open about the limitations and the true predictive capacity. The year 2012 is described as an anomalous year. How can it then be used for extrapolation in both time and space in the regression analysis and further in the text? The analysis should also use published data for validation and check the spatial validity of the algorithms. Probably they are very limited and cannot be extrapolated to any other region. RMSE are large for DIC

**and TA, what consequences do that have for the calculated $\Omega$ ? Explain.**

- --As mentioned below, errors in  $\Omega$  resulted from errors in analytical methods, select of constants, difference between observed and estimated have been estimated. Cumulative errors in estimated values will be indicated in Figures in the revised manuscript.
- *3)* The uncertainties in the calculations of the anthropogenic CO2 impact is not clearly described and emphasized in the abstract without a solid ground.
- -- We will indicate uncertainties in Figure 7. The sentence was removed from the abstract. Abstract has been modified as shown below.

**More specifically:**

Title: Since 2013 did not show undersaturation, also the large range of RMSE also show "prolonged" oversaturation! The title is biased towards showing undersaturation and overstates the "negative" results and does not mirror the large uncertainty. Suggest to add "possible" or focus in the 2012 event with largely undersaturated waters due to biological processes (likely).

--The title has been changed to:

"Seasonal variation of CaCO3 saturation state in bottom water of a biological hotspot in the Chukchi Sea, Arctic Ocean"

**Abstract:**

Include range of saturation instead of only mentioning undersaturation

Row 16-: Highly speculative statement that it is anthropogenic CO2 uptake that drives the duration of CaCO3 saturation in the bottom waters. See also later comment. The authors mention the limitations of the method they used in the actual chapter which is not the case in the abstract. Rephrase this part to give possible rates of change per decade at a moderate scenario. RC8.5 is considered unrealistic.

-- The abstract has been changed to read:

"Distribution of calcium carbonate saturation state ( $\Omega$ ) was observed in bottom waters of the Chukchi Sea in autumn 2012 and early summer 2013. Aragonite and calcite undersaturation with  $\Omega$  as low as 0.3 and 0.5, respectively, were found in high productivity regions in autumn 2012 but not in early summer 2013. Comparison with other parameters has indicated that biological processes -respiration and photosynthesis- are major factor controlling regional and temporal variability of  $\Omega$ . From these ship-based observations, we have obtained empirical equations to reconstruct  $\Omega$  from temperature, salinity and apparent oxygen utilization. Using two-year-round mooring data and these equations, we have reconstructed seasonal variation of  $\Omega$  in bottom water in Hope Valley, a biological hotspot in the southern Chukchi Sea. Estimated  $\Omega$  was high in spring and early summer, decreased in later summer, and kept relatively low in winter. Calculations indicated a possibility that bottom water could have been undersaturated for aragonite on an intermittent basis even in the preindustrial period, and that anthropogenic CO2 has extended the period of aragonite undersaturation to two- or three-fold longer by now."

**Introduction:**

- Row 16 to 17: This is not correct. Just because it is a shallow shelf it does not mean that the Chukchi Sea has large content of anthropogenic CO2. One of the largest content of anthropogenic CO2 is found in the North Atlantic, which is several thousand meters deep. Use a solid reference if that is the case, if not, delete sentence.
- --We do not mean Chukchi Sea bottom water has a large content or higher content of anthropogenic  $CO_2$  than in the North Atlantic but mean that anthropogenic  $CO_2$  can penetrate easily to bottom water by mixing of surface water into bottom water induced by wind, tide, and atmospheric cooling.

We modified the sentence to:

"because of shallow bottom depth of  $\sim 50$  m, vertical mixing induced by wind, tide or atmospheric cooling brings anthropogenic CO2 into bottom water to which benthos are exposed."

**2. Observation and analysis**

**Generally, lack of detailed information on methodology**

Row: 5 to 7. Two different methods are used to determine TA, spectrophotomery and potentiometric titration. How did these two methods differ? Was this assessed? The quality of the TA data plays a large role for the determination of the CaCO3 saturation. Is there any comparison performed between these two methods? Or was another parameter measured for example pH, as to perform internal consistency check on the quality of the TA and DIC? This information would greatly support some of the findings later in the manuscript and also strengthen the algorithm development.

--We do not have pH or other measurements. Intercomparison of two method showed good agreement of 0.88±2.03 umol/kg (Li et al., 2013). Also, a comparison of S-TA relationship for our two cruises did not show any offset. These are now mentioned in the text.

- Row 5: How was dissolved oxygen measured on the ship? What was the method for nutrients? If they were sensor data should also include information on what sensors and how and when they were calibrated. Perhaps the difference on DO in bottom water (Figure 2) between 2012 and 2013 only due to sensor differences? That should be explained.
- --Bottle DO was determined by Winkler titration following World Ocean Circulation Experiment Hydrographic Program operations and methods (Dickson, 1996). Nutrient samples were analyzed according to the GO-SHIP Repeat Hydrography Manual (Hydes et al., 2010). These are now mentioned in the text.
- Row 8 to 10: The precision for DIC is quite high, was there a reason for that? How much will that error result in calculated  $\Omega$ ?
- -- We do not know the reason for relatively high precision for DIC in 2013. Uncertainty of 5.5 umol/kg in DIC results in error of  $\pm 0.05$  in calculated  $\Omega$ .
- Row 12: Why was Lueker constants chosen? It is preferable if the authors would present results from several other determinations of constants, estimate the mean for all and deviation from the mean for each constant as to assess the range in uncertainty/error by using different set of constants.
- --This kind of analysis has been done elsewhere (Azetsu-Scott et al., 2010) and we do not think this is worth to repeat the similar analysis in our manuscript. We have used Lueker following our previous publications. When constants of K1, K2 from Mehrbach et al., 1973 (refit by Dickson and Millero, 1987) were used,  $\Omega$  differs from original estimate by only < 0.01.

Row 17: Describe the sensors (brand), resolution and how the accuracy was established. Row 17: Was the sensor measuring chlorophyll- a or fluorescence? What sensor was used? Was it calibrated?

-- We have added description about DO sensor as follows:

"DO sensor used for mooring observation was an AROW-USB phosphorescent DO sensor (JFE Advantech Co., Ltd., Kobe, Japan). The sensor was calibrated using oxygensaturated and anoxic water to determine the linear relationship between them with 2 % accuracy (Nishino et al., 2016). .....difference between sensor and bottle DO data was 4 umol/kg." Because we do not use chlorophyll a or turbidity data, we do not think detailed information of these sensor and its calibration is needed to be mentioned in our manuscript. They are described in Nishino et al. (2016) as referred in the text.

**3.1 Ship based observations**

Since the Figure does not refer to any of the sampling locations such as (Pt Barrow or Hope valley) it is difficult to follow. Please add this information in the figure referred to in the text. The authors should also show the TA and DIC not only CaCO3 saturation. The discussing that follows on the differences between the two years would greatly benefit from analysis of the TA and DIC data.

--Revised as suggested.

**3.1 Ship based observations**

Row 18 to 20: Section is repetitive and is almost the same as stated in introduction, redundant.

--The sentence has been deleted.

- Row 24: Interesting that undersaturation was found in the Bering Strait. This was not found in 7-years earlier (water column data summer 2005 by Chierici and Fransson (2009). Could that support the statement that undersaturation has progressed in the area? Interesting comparison to add. They also discuss different constants and the result in  $\Omega$  (see previous comment).
- --Their observations in Bering Strait was made in July/August. Therefore, we think that the difference between observations reflects seasonal variation of  $\Omega$ . As shown above, use of different constant cannot produce a large difference in  $\Omega$  as observed between cruises.
- Row 32-end of section: The authors explain the difference between the two years of shipboard measurements to be caused by differences in organic matter accumulation. Was there evidence for larger primary production in 2012 than in 2013? Perhaps sediment trap data? Or satellite data showing differences in primary production? Later in the manuscript the authors refer to Grebmeier et al 2015 for chlorophyll a data. This could be developed further.
- --As mentioned in the text, we think that stronger stratification in 2012 caused an accumulation of more CO2 in the bottom water because photosynthetic activity was lower in 2012 than in 2013 associated with stronger stratification and mixing with sea

ice meltwater (Nishion et al., 2016).

**3.2: Mooring observations**

Row 3: Display that sensor data agreed well, how well? Show some numbers?

- --Differences between bottle data and sensor data for the day of bottle sampling will be indicated in a table as suggested by the reviewer. Difference between sensor DO and bottle DO of 4 umol/kg in July 2013 is mentioned in the text.
- Row 6: what does the author mean by" T, S, DO showed larger and high frequent variability"?
- --We meant that these sensor data showed a lot of ups and downs. We have modified the sentence to:
- "T, S and DO showed large fluctuation"
- The changes in water masses limit the possibility to use algorithms to estimate aragonite saturation in this area, since it is highly variable. Should add information on the limitations of the spatial extent for the algorithms.
- --We believe that the obtained equations are applicable for waters in the eastern Chukchi Sea as confirmed by comparison of estimates with observations for multiple cruises. If processes affecting  $\Omega$  as well as source of waters are the same, the same equation could be applicable in the western Chukchi Sea. However, we do not have any data to validate this.
- Row 11-12: What about the ice conditions between the two years? Could variability in sea ice formation/melting cause these changes? Did they differ between the years?
- --Winter conditions are similar between two years. More sea ice meltwater was found in summer of 2012. This is stated in the text as the cause of stronger stratification and accumulation of more CO2 in bottom water in 2012.

3.3 Regression analysis

Generally: This chapter is not open on its limitations, and does not give a ground-based insight into the development of algorithms or the predictive power of the proxies. This requires a much more thorough and detailed and unbiased description of the methodology that is used here and its limitations. For example: What is the cumulative error in the regression analysis including all error (analytical methods, CO2SYS calculation, difference between observed and estimated). The Figures where this data is used should all include error bars.

- Figure 6: The errors and biases are large as is shown in the RMSE (grey lines). This error "only" includes the error from reconstruction and do not include the cumulative error from the  $\Omega$  calculation from DIC and TA. The figure text should include the full regression incl coefficient of correlation.
- --As mentioned above, errors in  $\Omega$  resulted from errors in analytical methods, select of constants, difference between observed and estimated have been estimated. Cumulative errors in estimated values will be indicated in Figures in the revised manuscript.

*Row 15: DIC is also influenced by mixing of water masses, not only TA.* --This sentence has been deleted.

- *Row 17: What about calcification by primary producers in the area? That could also cause changes in the TA.*
- --This possibility is mentioned in the revised text:
- "A bloom of calcifying primary producer can cause a drawdown of TA (Murata et al., 2002). However, neither bloom of calcifies nor TA drawdown in S-TA diagram was observed during our observations."
- *Row 19: the authors should include the TA –S figure. This will add information on the scatter in the data.*
- --Revised as suggested.
- Why was the chl a not used? Chl a is commonly used as proxy for primary production. AOU could be a proxy for respiration. Check also with fCO2 which generally correlates well with DO. If they do not correlate that could help in explaining other causes for differences. Explain the choices better.

--Chl a is a proxy for primary production at the moment of observation, while AOU represents the effect of primary production/respiration accumulated for some period. The latter is also the case for DIC. Therefore, AOU is a better proxy for effect of respiration/production on DIC. In fact, chl a and fCO2 does not show significant correlation (r=0.12, p=0.08) whereas AOU correlates well with fCO2 (r=0.92, p<0.01). A figure showing AOU-fCO2 correlation has been added to the revised manuscript.

Row 20: First the authors mention that the best predictions were obtained using all three

proxies, then they end up using fewer and fewer, this is understandable since it is better to be able to estimate the whole carbonate system as simple as possible. But it is not convincing and do not explain the selection of proxies. Most published algorithm development use a stepwise method to go forward with the proxy selection.

- --We have compared results of regression analysis and found the inclusion or replacement of proxies of biological processes, such as nutrients or chlorophyll a concentration did not improve the estimate of DIC and TA. This is now mentioned in the revised text.
- Moreover, the RMSE for TA is about 14 µmol/kg, which is quite substantial. What is the consequence for the calculated  $\Omega$ ? Need also to consider the RMSE i DIC which is even larger (24 µmol/kg). The authors need to address what that means for the calculated  $\Omega$  values.
- --Resulting error in  $\Omega$  is indicated in the comparison between estimated and observed  $\Omega$  in the text and in Figure 5a.
- Row 30: the RMSE for aragonite and calcite are quite large and cannot be neglected. The authors should perform a proper error analysis and estimate the cumulative error and discuss the error in context of the seasonal and interannual differences they found. This is crucial for the discussions after on trends and future projections.
- --As mentioned above, errors in  $\Omega$  resulted from errors in analytical methods, select of constants, difference between observed and estimated have been estimated. Cumulative errors in estimated values will be indicated in Figures in the revised manuscript.

**Row 29: What is the cause for the large difference in the surface waters? Explain What is the RMSE for the fits mentioned here?**

--Large differences were found in surface waters with high temperature (>6°C). This might be due to rapid warming at the surface that could cause decoupling of oxygen and carbon due to a difference in temperature dependence of solubility or in gas exchange rate between two gases. We mention these possibilities in the revised text. We do not estimate RMSE for these data because scattering of these data is evident in Figure 5b and because they are out of range of bottom water in temperature and  $\Omega$ .

Row 32. It is repeatedly displayed in the results between the two years that this is a highly variable and dynamic area. This also means that the algorithms are likely to be

highly uncertain and result in high bias for other times of the year and also between years. Thus, using the algorithms for 2000, 2002, 2006, 2009, and 2010 is highly speculative. The authors states that this is a validation of the method but to be a validation the authors need independent data on shipboard DIC and TA to compare with the estimated DIC and TA from the algorithms. This is a great weakness in the method and I think the authors should remove this analysis since it is highly speculative. Especially since the authors later on uses results from this analysis to make projections on future saturation states. There are likely other published DIC and TA data in this region which could be used for a proper validation. This area is one of the most studied in the Arctic Ocean with regard to the carbonate system.

- Moreover, using sensor data from more than 10 years must give details on sensor types, were they the same for all those years? Calibrations? Precision? Drift?
- --We found that the concern of the reviewer is based on misunderstanding. We have validated the algorithms using independent data of shipboard DIC and TA with the estimated DIC and TA from the algorithm. Please see our reply above.
- Page 7: Row 1-4: the authors mention that the estimated and observed omega values correlate well, RMSE is 0.36 and 0.57!! this give a very large relative standard deviation (CV%) compared to the mean Ω, and is not what should be referred as "well correlated". Again, needs a proper error analysis and put into context of the seasonal and interannual variability in the omega values. Be more objective.
- --Because  $\Omega$  is a ratio, relative standard deviation CV% does not make sense. However, we agree that we should be more objective in describing the results. We have removed the word "well" from the sentence.

**3.4.**

- Page 8: Row10: If 2012 was such an unusual year (or autumn), how can the algorithm that is developed on this data give reliable estimates for other years (and other areas)? Sometimes 10 years back in time? Not solid and again shows that the data from this study area cannot be extrapolated to other times of the year nor to other years.
- --See our response above.

**3.5 Anthropogenic impact**

The used method assumes that the rate of uptake in the bottom water is the same as in the surface water uptake of anthropogenic CO2. That should be added to the limitations of this method. The method also assumes that ocean mixing and all other processes have remained the same, which is partly commented by the authors regarding biological processes. It could be worthwhile using reported values of decadal uptake rates from for example Tanhua et al., 2007 Anthropogenic CO2 in the Arctic Ocean) and use that rate of change to estimate the impact of anthropogenic uptake. That is more robust and also shows the rate of change at depth, which is likely not the same as in the surface waters. Regardless the authors should include more on the assumptions and limitations on the resulting omega values.

--Estimates of anthropogenic CO2 by Tanhua et al. (2009) are basically the same as estimated in our study for waters above the winter mixed layer, which is the bottom depth for the Chukchi Sea. TTD method is accounting for mixing of water with different ages, but such mixing does not significantly occur in our study area.

Assumptions that processes have remained the same since preindustrial period is not probable and therefore our estimate is rough.

This is now stated in the text as: "In order to roughly estimate the effect of anthropogenic CO2..." and "Caveat here is that our calculation is based on an assumption that terms  $\Delta$ diseq and  $\Delta$ bio have not changed since pre-industrial period and therefore provides only very rough estimates". Numerical number of counted days of undersaturation for preindustrial and future cases are not presented in the revised manuscript.

- Also, after the last Paris (COP) meeting it was agreed that the RCP8.5 scenario is unrealistic and I suggest that the authors include a moderate scenario in addition.
- --In the revised manuscript, we removed the expression of "50 years" for pCO2=650ppm case.
- Also, now it is difficult to follow the actual change in omega, it would be better to suggest a rate of change /decade. How much would omega change due to anthropogenic CO2 per decade?
- --A rate of change in omega /decade differs significantly depend on revel factors that varies during a course of a year. This is shown in Figure 7, as the fact that difference between preindustrial and future cases varies with seasons. Describing a rate of change/decade for each season will beyond the limitation of our rough estimate.
- *Figure 7: Perhaps combine Figure 6 and Figure 7 to show present, pre-ind and a future 50 years.*
- --We did not combine them as 3 estimates with error bars make the figure too busy.

**Figure 8: Redundant picture. Also, Figure caption text is wrong, should probably be "collected".**

--We do not think this a redundant picture. It supports the statement in the text that many bivalves were found in Hope Valley hot-spot area, both well-grown adults and small young individuals. However, because a couple of Figures have been added in the revised manuscript as suggested by the reviewer, it should be better to reduce the number of figures. We therefore moved this picture to Figure 7 as an insert panel.

**All of following comments were considered and manuscript was revised accordingly**

Rewrite summary with regard to previous comments.

Biological hotspot is used many times.

Reference spelling! Kroeder!?

Figure 1: Please show the main study sites on the map Pt Barrow and Hope Valley, Bering Strait.

Please point out where the mooring was located in Figure 1.

Figure 2: Figure 2c: add abbreviation DO which is used in the text and unit for DO.

Add plots showing DIC, TA and fCO2. Include units. Moreover, there is no undersaturation in 2013 which is not the impression given in the text. Please, clarify in the text.

Figure 3: Describe what a negative and positive AOU refers to.

Figure 4: Difficult to see the bias between shipboard and sensor data in the plot, suggest to add a Table showing the differences. The Table should also include the number of samples/data points that were compared (N).

Figure 5a: Show the data from 2012 and 2013 with different symbols or colors. Figure 5b (xaxis), is the Ωobs based on 2012 and 2013 data? Clarify. Are the displayed data mean values for all years between 2000-2010 or separate years? If separate years, show the different years using different symbols for improved understanding of the bias to the observed for the different years.

---

## Author Response (AR1)

**Reply to comments of Reviewer #1, Professor Lee Cooper**

**We would like to thank Professor Lee Cooper for a thorough review of this paper. We have addressed all comments and suggestions. Please find our responses below**

*I have more misgivings about the extension of using the moored data and apparent correlations developed between AOU and ocean acidification to estimate undersaturation of calcite minerals over the course of the year. While I am not surprised that undersaturation is probably common due to mineralization and high productivity, the conclusions are based upon the assumption that oxygen utilization continues at fairly constant rates over the winter, and I think the small published set of*

10 *sediment oxygen utilization measurements available from arctic shelves does not strongly support this assumption. Only one study (Devol et al. 1997) is cited to support this assumption, and it sampled in the winter in unproductive waters much different from the moored site.*

**--We agree that we need to discuss more about the winter AOU and its correlation to our reconstruction of CaCO₃ saturation state. A relatively constant positive AOU (~50 µmol kg⁻¹) was observed over the winter for two years. As**

15 **mentioned in the text, same level of AOU was also found in the hypersaline water that is formed in contact with atmosphere. This suggests that positive AOU in winter bottom water is not due to insufficient gas exchange but oxygen consumption. Although there is no year-round observation of sediment oxygen uptake in southern Chukchi Sea, it is known that oxygen uptake rate has a seasonal variation and is low in winter prior to initiation of biological production in spring. Previous studies in other Arctic waters have observed that sediment oxygen uptake rate in winter is not zero**

20 **but is about half of that in summer in coastal area north of Pt. Barrow, Alaska (Devol et al., 1997), in Young Sound in Northeast Greenland (Rysgaard et al., 1998) and in Resolute Bay in Canadian high Arctic (Welch et al., 1997). Winter AOU observed by moored sensor in our study was about 1/3 ~ 1/2 of autumn AOU. Therefore, we presume that the positive AOU in bottom water during winter can be explained by continued sediment oxygen uptake. Discussions on this have been included in the revised text as follows:**

25 **(p7, line 2-) "Continued sediment oxygen uptake is a possible reason for the undersaturation. Previous studies in shallow Arctic seas have found that sediment oxygen uptake rate is regulated by the availability of organic matter and macrofaunal biomass (Grebmeier and McRoy, 1989; Rysgaard et a l., 1998; Grant et al., 2002; Clough et al., 2005). Accordingly, oxygen uptake rate has a seasonal variation and is low in winter prior to initiation of biological production in spring (Cooper et al., 2002; Grant et al., 2002). Nevertheless, winter sediment oxygen uptake rate is not zero but is**

30 **about half of that in summer in coastal area north of Pt. Barrow, Alaska (Devol et al., 1997), in Young Sound in Northeast Greenland (Rysgaard et al., 1998) and in Resolute Bay in Canadian high Arctic (Welch et al., 1997). In our observations, mean AOU in mid-winter (February to April) was 1/3 ~ 1/2 of that in October. Therefore we consider that positive AOU was maintained by benthic respiration during winter.**

**(p9, line 14-)**" Although equations obtained from summer/autumn data were used to estimate winter Ω, we presume this is acceptable because summer/autumn bottom water is a remnant of winter water that was modified by remineralization of organic matter after spring. If remineralization quotient of DO and DIC is held relatively constant over the course of the year as observed in Young Sound (ΔDIC/ΔDO = 0.8~1.1, Rysgaard et al., 1998), the summer/autumn relationship between DIC (and TA) and T, S, AOU could be applicable to winter data. This assumption should be verified by direct winter observation by ship-based sampling, chemical sensors, or automatic water samplers in the future."

*The moored data used (Nishino et al. 2016) also had to be managed—corrections undertaken for AOU data that were corrected because of apparent issues with the data that are mentioned in Nishino et al. 2016.*

--We have carefully checked data and calculations and found that there was a mistake in unit conversion (from saturation % to μmol/kg) in Nishino et al. (2016) for the 3$^{rd}$ set of mooring data. This was the cause of the large difference of 69 μmol/kg between bottle and sensor measurements mentioned in their paper. With correct unit conversion, the difference was only 4 μmol/kg. In our original manuscript, we did not use calculations by Nishino et al. (2016) and used data correctly converted from original sensor output. Therefore, this mistake does not affect our results. This issue is now mentioned in the text to not cause same concern to readers:

**(p4, line 17-)** "Note that a correction of -69 μmol kg$^{-1}$ applied for the third mooring data in Nishino et al. (2016) was found to be due to an artificial error in conversion of original sensor output to concentration in μmol kg$^{-1}$. With correct conversion, the difference between sensor data and bottle data obtained on 1 September 2013 was reduced from 69 μmol kg$^{-1}$ to 4 μmol kg$^{-1}$. Accordingly, we did not apply any correction to DO sensor data in the present paper."

*Finally, the use of this correlation method for estimating calcium carbonate dissolution potential was initially demonstrated in California and Oregon, so it really hasn't been confirmed to work in the Arctic where there are much more extreme seasonal changes in biological activity.*

--Our study is the first attempt to use this method to highly productive Arctic shelf sea. We agree that this should be confirmed in the future by using direct observations of carbonate parameters throughout the year. We have noted this in the revised text to read:

**(p11, line 30-)** "We should note that this study is the first attempt to reconstruct seasonal variation of Ω using a method that has not been confirmed to work in Arctic shelf seas where seasonal changes in biological activity are extremely large. Direct observation of carbonate parameters in winter by using sensors or water sampler is desired to confirm our results.

*The authors defend their approach by stating that their shipboard sampling bracketed both high productivity in July and high oxygen utilization in October although my examination of the Nishino et al 2016 results suggest that sampling in July may have missed the highest primary productivity.*

**--We agree that the maximum chlorophyll a was observed in May/June. However, our two shipboard samplings were made in two different period with high and low DO conditions. To be more accurate, text has been modified from "ship-based observations captured both higher and lower ends of seasonal variation in DO" to (p6, line 14-) "ship-based observations were made when DO was at higher and lower parts of seasonal variation", from "our ship-based observations in autumn 2012 and summer 2013 have captured the lowest and the highest $\Omega$ periods, respectively." to (p9, line 7-) "our ship-based observations in autumn 2012 and summer 2013 have captured low and high $\Omega$ periods, respectively."**

*I don't think this is necessarily a flawed paper because the available evidence suggests that widespread undersaturation with respect to carbonate minerals on productive arctic shelves is probably correct, but I don't think the evidence provided here is strongly convincing either.*

**--We have revised the manuscript to describe results in an honest manner. Sentences have been changed to be more exact and fair, for example, "bottom water was kept at aragonite undersaturation for most of the winter" was changed to "$\Omega$ in bottom water was kept low during winter" and "intermittent undersaturation was found" was changed to "intermittent undersaturation was suggested".**

**The title was also change from "prolonged undersaturation…" to "Seasonal variation of $CaCO_3$ saturation state in bottom water of a biological hotspot in the Chukchi Sea, Arctic Ocean".**

**We hope the revised manuscript will meet the requirements for publication.**

*The manuscript is also unevenly written, and would benefit from efforts of a native English language editor. A number of mistakes in spelling, in the references, and even in the spelling of the author names suggest a hasty assembly of the manuscript. I have provided some editing suggestions below and posed a few additional questions and concerns, but this is not a comprehensive editing effort.*

**--We are sorry that we have submitted the manuscript with many typographical errors and really appreciate your kind editing. We will ask an English Language Service to edit our revised manuscript when it is accepted for publication.**

*Page 2: Line 25. I don't follow why the reference to Talmange and Gobler, 2009 needs to be made here. This reference has already been made (prior page, Line 29) to document larval stage vulnerability, although that reference is about non-polar invertebrates. The statement and reference repeated here is redundant*

**--The reference has been removed. The sentence has been modified to read (p2, line 10-) "Because many benthic organisms have planktonic larval stages, timing and duration of CaCO₃ can be critical for their growth and populations".**

5   *Page 5: Line 27-30. Most of the published data for sediment oxygen utilization rates for the northern Bering and Chukchi seas indicates that there is significant seasonal variation and it is lower in the late winter prior to initiation of the sea ice edge bloom. I think the Devol et al. paper is dubious to cite here because the winter sampling was done in nutrient-poor, near-shore waters that do not have high AOU at any time of year.*

**--See response to the first comment above.**

*Page 8. Line 16. This really isn't a complete sentence.*

**--The text has been changed as follows:**

**(p 10, line 9-) "In order to roughly quantify the effect of anthropogenic CO2 on timing and duration of CaCO3 undersaturation in our 2-year time series of Ω, we have estimated Ω in two cases: 1) preindustrial period case with**

15   **atmospheric partial pressure of CO2 (pCO2) of 280ppm, and 2) future case with pCO2 of 650 ppm. Following previous studies (Gruber et al., 1996; Sabine et al., 1999; Yamamoto-Kawai et al., 2013; 2015), DIC concentration in year t is expressed as: DICt = DICEQt-0 + (Δdiseq + Δbio), where…".**

*Page 9. Line 6-7. The sentence is not grammatically correct and I am not sure what the authors are trying to say.*

20   **--The sentence has been deleted.**

*Page 9. Line 25. Change "to" to "from"*

**--We could not find "to" in this line and are not sure where the reviewer found this error. This will be corrected when we have the professional English editing.**

*Page 9. Lines 27-29. The Nishino et al. 2016 paper appears to show that the maximum chlorophyll a bloom can occur prior to July, so the early summer sampling may not have successfully sampled the most productive period.*

**--We agree that chlorophyll a was highest in June, though it is still high in July. The text has been changed from "under photosynthesis in early summer 2013" to (p11, line 19-) "under an influence of photosynthesis".**

**All of following have been changed as suggested.**

*Page 1: Line 3. I think Dr. Nishino's name is misspelled.*

*Page 1: Line 26. Change "to affect" to "which affects"*

*Page 2: Line 3. Salisbury et al. reference misspelled.*

*Page 2: Line 7. Change "Nutrients. . ..is carried" to "Nutrients. . ..are carried"*

*Page 2: Line 8. Change "making the sea to have very high primary productivity" to "promoting very high primary productivity"*

*Page 2: Line 9. Add the article "A" before "proportion"*

*Page 2: Line 14. Change spices to species*

5 *Page 2: Line 22. Lower case 3 needed for calcium carbonate molecular symbol.*

*Page 2: Line 24. Change "difficulties in" to "the lack of"*

*Page 3: line 2. Change was to were*

*Page 3: line 15. "the" before maintenance not necessary*

*Page 3: line 27. Delete "that" and change "is" to "as"*

10 *Page 3: Line 29. Insert a "the" before "two visits"*

*Page 4: Line 10. Change "kept at near" to "remained at a near"*

*Page 5: Line 23. Change kept to remained*

*Page 5: Line 31. Change captured to sampled*

*Page 6: Line 26. Change "in" to "to a"*

15 *Page 7: Line 15. Remove "of" The sentence would also read better if it starts with the article "the"*

*Page 7: Line 25. Suggest should be suggests.*

*Page 7: Line 28. Devol reference should be 1997, not 1996.*

*Page 7: Line 30 persisted should be persistent.*

*Page 8. Line 31. Change "process" to "processes" and "is" to "are"*

20 *Page 9. Line 1. Change is to are*

*Page 9. Line 2. Add "the" between that and primary production.*

*Page 9. Line 3. There is a Grebmeier, 2012 reference in the literature cited, but not a Grebmeier et al. 2012.*

*Page 9. Line 9. Change "even with half productivity than today" to "even with half the productivity occurring today"*

*Page 9. Line 15. Change "it is indicated" to "it suggests"*

25 *Page 9. Line 16. Change "occupation" to "the proportion of"*

*Page 9. Line 17. Change "occupies" to "increases to"*

*Page 9. Line 18. Change These to This and indicate to indicates; add the article "a" has and significant.*

*Page 9. Line 22. I suggest changing Horizontal to Spatial*

*Page 9. Line 26. The mooring observations are presented in Nishino et al. 2016, so I think it is more accurate to state that the*

30  *authors used the data from Nishino et al. 2016 to estimate calcium carbonate undersaturation.*

*Page 9. Line 33. Change "Occupation of calcite" to "The period of calcite"*

*Page 10, Line 2. Insert "subject to" between "been" and "aragonite"*

*Page 10. Line 5. Change two-hold to two-fold; change "occupation" to "the period of"*

*Page 10. Line 6. Change "year-long occupation under highly stratified condition.*

*Occupation. . ." to "year-round periods under highly stratified conditions. Periods of"*

*Page 10. Line 8. I suggest changing "surely" to "clearly". It is less colloquial and more specific*

*Page 10. Line 10. Kroeker's name is misspelled.*

*Page 10. Line 12. Change "may be conflicting the fact" to "is not consistent with the fact"*

5    *Page 10. Line 29. Since there was no formal presentation of oxygen isotope data, I don't think an acknowledgement is necessary.*

*Page 12. Line 17. Global Change Boil should be Global Change Biol*

*Page 13, line 25. Raven reference is not in alphabetical order.*

*Figure 1. The arrows identifying the mooring sites are not clear.*

10    *Figure 8 caption. Corrected should be collected. Also trawl is misspelled.*

[revised manuscript text omitted]

**Fig. 5̶8**: Comparison between Ωar estimated from T, S, and AOU̶, data using equation (1) and (2) (Ωar o̶b̶s̶e̶r̶v̶e̶d̶ ̶d̶u̶r̶i̶n̶g̶(est)), and Ωar estimated from bottle DIC and TA (Ωar (obs)) for ship-based cruises (a) in 2012 and 2013, and (b) in 2000, 2002, 2006, 2009 and 2010.

[Figure]

Fig. 6: Time series of Ωar (top) and Ωca (bottom) reconstructed from mooring data of T, S and AOU (black line). Red symbols indicate ship-based observations. Gray lines indicate a range of -0.36 to +0.45 for Ωar and -0.57 to +0.71 for Ωca (see text) .

[Figure]

**Fig. 9: Time series of Ωar (top) and Ωca (bottom) reconstructed from mooring data of T, S and AOU. Ω values (black line) are shown with range of uncertainty (grey lines).**

[Figure]

**Fig. 10**: Time series of $\Omega ar$ (top) and $\Omega ca$ (bottom) for cases when atmospheric $CO_2$ concentration was 280 ppm (blue; pre-industrial period) or 650 ppm (red; 50 years later). See text for details.

[Figure]

Fig. 8: Photo of bivalves corrected by a dredge taw near the mooring location during T/S Oshoro-Maru cruise in 2013.